# REGENT: A Retrieval-Augmented Generalist Agent That Can Act In-Context in New Environments

**Kaustubh Sridhar**[1]**, Souradeep Dutta**[1,2]**, Dinesh Jayaraman**[1]**, Insup Lee**[1]
[1]University of Pennsylvania, [2]University of British Columbia
`ksridhar@seas.upenn.edu`

## Abstract

Building generalist agents that can rapidly adapt to new environments is a key challenge for deploying AI in the digital and real worlds. Is scaling current agent architectures the most effective way to build generalist agents? We propose a novel approach to pre-train relatively small policies on relatively small datasets and adapt them to unseen environments via in-context learning, without any fine-tuning. Our key idea is that retrieval offers a powerful bias for fast adaptation. Indeed, we demonstrate that even a simple retrieval-based 1-nearest neighbor agent offers a surprisingly strong baseline for today's state-of-the-art generalist agents. From this starting point, we construct a semi-parametric agent, REGENT, that trains a transformer-based policy on sequences of queries and retrieved neighbors. REGENT can generalize to unseen robotics and game-playing environments via retrieval augmentation and in-context learning, achieving this with up to 3x fewer parameters and up to an order-of-magnitude fewer pre-training datapoints, significantly outperforming today's state-of-the-art generalist agents. Website: https://kaustubhsridhar.github.io/regent-research

## 1 Introduction

AI agents, both in the digital [38, 19, 37, 28, 53] and real world [5, 7, 63, 33, 48, 24], constantly face changing environments that require rapid or even instantaneous adaptation. True generalist agents must not only be capable of performing well on large numbers of training environments, but arguably more importantly, they must be capable of adapting rapidly to new environments. While this goal has been of considerable interest to the reinforcement learning research community, it has proven elusive. The most promising results so far have all been attributed to large policies [38, 19, 37, 28, 5], pre-trained on large datasets across many environments, and even these models still struggle to generalize to unseen environments without many new environment-specific demonstrations.

In this work, we take a different approach to the problem of constructing such generalist agents. We start by asking: Is scaling current agent architectures the most effective way to build generalist agents? Observing that retrieval offers a powerful bias for fast adaptation, we first evaluate a simple 1-nearest neighbor method: "Retrieve and Play (R&P)". To determine the action at the current state, R&P simply retrieves the closest state from a few demonstrations in the target environment and plays its corresponding action. Tested on a wide range of environments, both robotics and game-playing, R&P performs on-par or better than the state-of-the-art generalist agents. Note that these results involve *no pre-training environments*, and not even a neural network policy: it is clear that larger model and pre-training dataset sizes are not the only roadblock to developing generalist agents.

Having thus established the utility of retrieval for fast adaptation of sequential decision making agents, we proceed to incorporate it into the design of a "Retrieval-Augmented Agent" (REGENT). REGENT is a semi-parametric architecture: it pre-trains a transformer policy whose inputs are not only the current state and previous reward, but also retrieved tuples of (state, previous reward, action) from a set of demonstrations for each pre-training task, drawing inspiration from the recent successes of retrieval augmentation in language modeling [20]. At each "query" state, REGENT is trained to prescribe an action through aggregating the action predictions of R&P and the transformer policy. By exploiting retrieval-augmentation as well as in-context learning, REGENT permits direct deployment

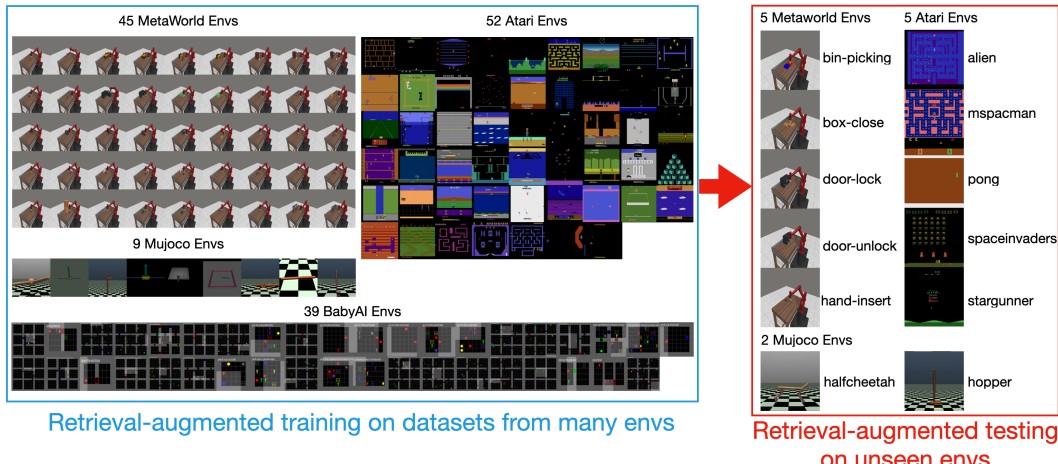

Figure 1: Problem setting in `JAT`/`Gato` environments.

in entirely unseen environments and tasks with only a few demonstrations. `REGENT` is only one of two models developed so far that can adapt to new environments via in-context learning: the other model is the multi-trajectory transformer (MTT) [37].

We train and evaluate `REGENT` on two problem settings in this paper, shown in Figures 1 and 2. The first setting is based on the environments used in `Gato` [38] (and its open source reproduction `JAT` [19]) and the second setting is based on the ProcGen environments used in MTT [37].

In both settings, `REGENT` demonstrates significant generalization to unseen environments without any finetuning. In the `JAT`/`Gato` setting, `REGENT` outperforms `JAT`/`Gato` even when the baseline is finetuned on demonstrations from the unseen environments. In the ProcGen setting, `REGENT` significantly surpasses MTT. Moreover, in both settings, `REGENT` trains a smaller model with 1.4x to 3x fewer parameters and with an order-of-magnitude fewer pre-training datapoints. Finally, while `REGENT`'s design choices are aimed at generalization, its gains are not limited to unseen environments: it even performs better than baselines when deployed within the pre-training environments.

## 2 RELATED WORK

Recent work in the reinforcement learning community has been aimed at building foundation models and multi-task generalist agents [38, 19, 5, 28, 7, 63, 33, 48, 24, 10, 22, 3, 17, 54, 40, 41, 55, 60].

**Many existing generalist agents struggle to adapt to new environments.** `Gato` [38], a popular generalist agent trained on a variety of gameplay and robotic environments, struggles to achieve transfer to an unseen Atari game even after fine-tuning, irrespective of the pretraining data. The authors attribute this difficulty to the "pronounced differences in the visuals, controls, and strategy" among Atari games. They also attribute `Gato`'s lack of in-context adaptation to the limited context length of the transformer not allowing for adequate data to be added in the context. Our method sidesteps this issue by retrieving only limited but relevant parts of demonstration trajectories to include in the context. `JAT` [19], an open-source version of `Gato`, faces similar problems. **We compare with and significantly outperform `JAT`/`Gato` using fewer parameters and an order-of-magnitude fewer pre-training datapoints.** While `REGENT` is a 138.6M parameter model, `JAT` uses 192.7M parameters. The closed-source `Gato`, with similar performance as the open-source `JAT`, reports using 1.2B parameters. `JAT` is also pre-trained on up to 5-10x the amount of data used by our method and yet cannot generalize to unseen environments. Even after finetuning on a few demonstrations from an unseen environment, `JAT` fails to meaningfully improve.

**Many recent generalist agents cannot leverage in-context learning.** In-context learning capabilities enable easier and faster adaptation compared to finetuning. Robocat [5], which builds on the `Gato` model, undergoes many cycles of fine-tuning, data collection, and pre-training from scratch to adapt to new manipulation tasks. The multi-game decision transformer [28], an agent trained on over 50 million Atari gameplay transitions, requires another 1 million transitions for fine-tuning on a

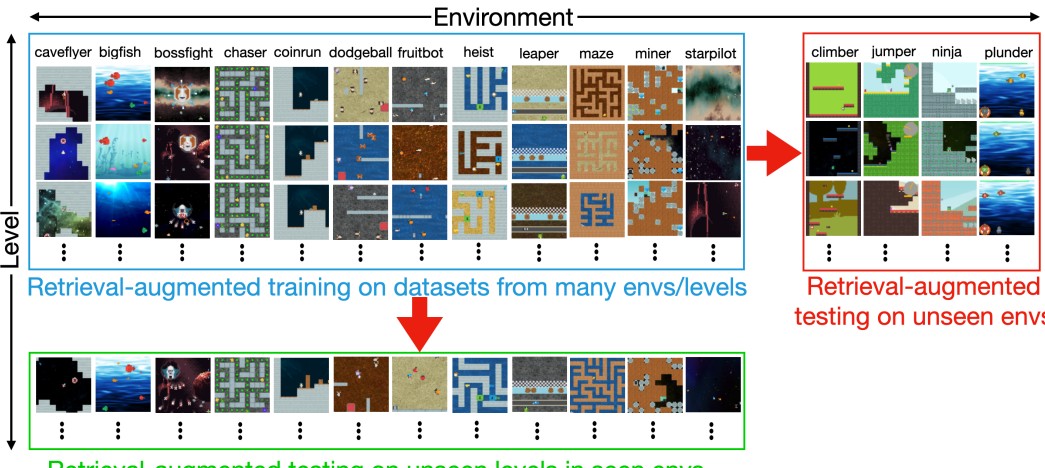

Figure 2: Problem setting in ProcGen environments adapted from [37].

held-out game which is not practical in real robot settings. We, on the other hand, show that `REGENT` (and even `R&P`) can adapt to new Atari games with as little as 10k transitions and no finetuning on said transitions. Finally, the RT series of robotics models [7, 63, 33], recent Vision-Language-Action models like Octo, OpenVLA, Mobility VLA [48, 24, 10], and other generalist agents like BAKU, RoboAgent [22, 3] do not evaluate or are demonstrated to not possess in-context learning capabilities. `REGENT` on the other hand can adapt simply with in-context learning to unseen environments.

**Agents that can adapt to new tasks via in-context learning, only do so within the same environment.** Algorithm Distillation [26], Decision Pretrained Transformer [27], and Prompt Decision Transformer [52] are three in-context reinforcement learning methods proposed to generalize to new goals and tasks within the same environment. None of these can handle changes in visual observations, available controls, and game dynamics.

**We also compare with and outperform MTT [37], the only other model that can adapt in-context in the ProcGen setting, with improved data efficiency and a smaller model size**. MTT trains a transformer on sequences of trajectories from a particular level and environment and adapts to unseen environments by throwing the demonstrations into its context. The ProcGen variant of `REGENT` use an order-of-magnitude fewer transitions in pre-training and is about one-third the size of MTT.

**Retrieval-augmented generation for training and deployment** is a core part of our policy. Various language models trained with retrieval-augmentation such as the original RAG model [29], RETRO [4], and REALM [21] have demonstrated performance on par with vanilla language models with significantly fewer parameters. Moreover, retrieval-augmented generation [20] with large language models has enabled them to quickly adapt to new or up-to-date data. We hope that our work can enable similar capabilities for decision-making agents.

## 3 PROBLEM FORMULATION

We aim to pre-train a generalist agent on datasets obtained from different environments, with the goal of generalizing to new unseen environments. The agent has access to a few expert demonstrations in these new environments. In this work, the agent achieves this through in-context learning without any additional finetuning. We assume that the state and action spaces of unseen environments are known.

We model each environment $i$ as a *Markov Decision Process (MDP)*. The $i$-th *Markov Decision Process (MDP)* $\mathcal{M}_i$ is a tuple $(\mathcal{S}_i, \mathcal{A}_i, \mathcal{P}_i, \mathcal{R}_i, \gamma_i, \mathcal{I}_i)$, where $\mathcal{S}_i$ is the set of states, $\mathcal{A}_i$ is the set of actions, $\mathcal{P}_i(s'|s, a)$ is the probability of transitioning from state $s$ to $s'$ when taking action $a$, $\mathcal{R}_i(s, a)$ is the reward obtained in state $s$ upon taking action $a$, $\gamma_i \in [0, 1)$ is the discount factor, and $\mathcal{I}_i$ is the initial state distribution. We assume that the MDP's operate over trajectories with finite length $H_i$, in an episodic fashion. Given a policy $\pi_i$ acting on $\mathcal{M}_i$, the expected cumulative reward accrued over the duration of an episode (*i.e.*, expected return) is given by $J(\pi_i) = \mathbb{E}_{\pi_i}[\sum_{t=1}^{H_i} \mathcal{R}_i(s_t, a_t)]$.

We denote the expert demonstration dataset corresponding to the $i$-th (training or unseen) environment consisting of tuples of (state, previous-reward, action) as $\mathcal{D}_i = \{(s_0, 0, a_0), (s_1, r_0, a_1), \ldots, (s_{N_i}, r_{N_i-1}, a_{N_i})\}$. Let us assume that we have access to $K$ such training environments for $i$ from 1 through $K$, and unseen environments indexed from $K + 1$ through $M$. We assume that the datasets $\mathcal{D}_i$'s are generated by acting according to an expert policy $\pi_i^*$ which maximizes $J(\pi_i)$. The training environments ($i \leq K$) have a sizeable dataset of expert demonstrations, while the unseen environments have only a handful of them, i.e $|\mathcal{D}_i| >> |\mathcal{D}_j|$, where $j > K$.

# 4 REGENT: A RETRIEVAL-AUGMENTED GENERALIST AGENT

Simple nearest neighbor retrieval approaches have a long history in few-shot learning [50, 2, 49, 43, 9]. These works have found that, at small training dataset sizes, while parametric models might struggle to extract any signal without extensive architecture or hyperparameter tuning, nearest neighbor approaches perform about as well as the data can support. Motivated by these prior results in other domains, we first construct such an approach for an agent that can learn directly in an unseen environment with limited expert demonstrations in Section 4.1. Then, we consider how to improve this agent through access to experience in pre-training environments, so that it can transfer some knowledge to novel environments that allows it to adapt even more effectively with limited data in Section 4.2.

## 4.1 RETRIEVE AND PLAY (R&P)

This is arguably one of the simplest decision agents that leverages the retrieval toolset for adaptation. Given a state $s_t$ from an environment $j$, let us assume that it is possible to retrieve the $n$-nearest states (and their corresponding previous rewards, actions) from $\mathcal{D}_j$. We refer to this as the context $c_t \in \mathcal{C}_j$. The set $\mathcal{C}_j$ is the set of all such contexts in environment $j$. We also call the state $s_t$ the query state following terminology from retrieval-augmented generation for language modeling [20].

The R&P agent takes the state $s_t$ and context $c_t$ as input, picks the nearest retrieved state $s'$ in $c_t$, and plays the corresponding action $a'$. That is, $\pi_{\text{R\&P}}(s_t, c_t) = a'$. We describe the retrieval process in detail later in Section 4.2. Clearly, R&P is devoid of any learning components which can transfer capabilities from pre-training to unseen environments.

## 4.2 RETRIEVAL-AUGMENTED GENERALIST AGENT (REGENT)

To go beyond R&P, we posit that if an agent learns to meaningfully combine relevant context to act in a set of training environments, then this skill should be transferable to novel environments as well. We propose exactly such an agent in REGENT. We provide an overview of REGENT in Figure 3 where we depict the reinforcement learning loop with the retrieval mechanism and the REGENT transformer. In the figure, we also include the retrieved context and query inputs to the transformer and its output interpolation with the R&P action. We describe REGENT in detail below.

REGENT consists of a deep neural network policy $\pi_\theta$, which takes as input the state $s_t$, previous reward $r_{t-1}$, context $c_t$, and outputs the action directly for continuous environments and the logits over the actions in discrete environments. In the context $c_t$, the retrieved tuples of (state, previous reward, action) are placed in order of their closeness to the query state $s_t$ with the closest retrieved state $s'$ placed first. Let $d(s_t, s')$ be the distance between $s_t$ and $s'$. We perform a distance weighted interpolation between the outputs of the neural network policy and R&P as follows,

$$\pi_{\text{REGENT}}^\theta(s_t, r_{t-1}, c_t) = e^{-\lambda d(s_t, s')} \pi_{\text{R\&P}}(s_t, c_t) + (1 - e^{-\lambda d(s_t, s')}) \sigma(\pi_\theta(s_t, r_{t-1}, c_t)) \quad (1)$$

$$\sigma(x) = \begin{cases} \text{Softmax}(x), & \text{if action space is discrete} \\ L \times \text{MixedReLU}(x), & \text{if action space is continuous} \end{cases} \quad (2)$$

where $\text{MixedReLU} : \mathbb{R} \rightarrow [-1, 1]$ is a tanh-like activation function from [46] and is detailed in Appendix A. Further, $L \in \mathbb{R}$ is a hyperparameter for scaling the output of the neural network for continuous action spaces after the MixedReLU. We simply set both $L$ and $\lambda$ to 10 everywhere following a similar choice in [46]. The function $\pi_\theta$ is a causal transformer, which is adept at modeling relatively long sequences of contextual information and predicting the optimal action [29, 4, 8, 28]. All distances are normalized and clipped to $[0, 1]$ as detailed in Appendix A. For discrete action spaces, where the transformer outputs a distribution over actions, we use a softened version of $\pi_{\text{R\&P}}$

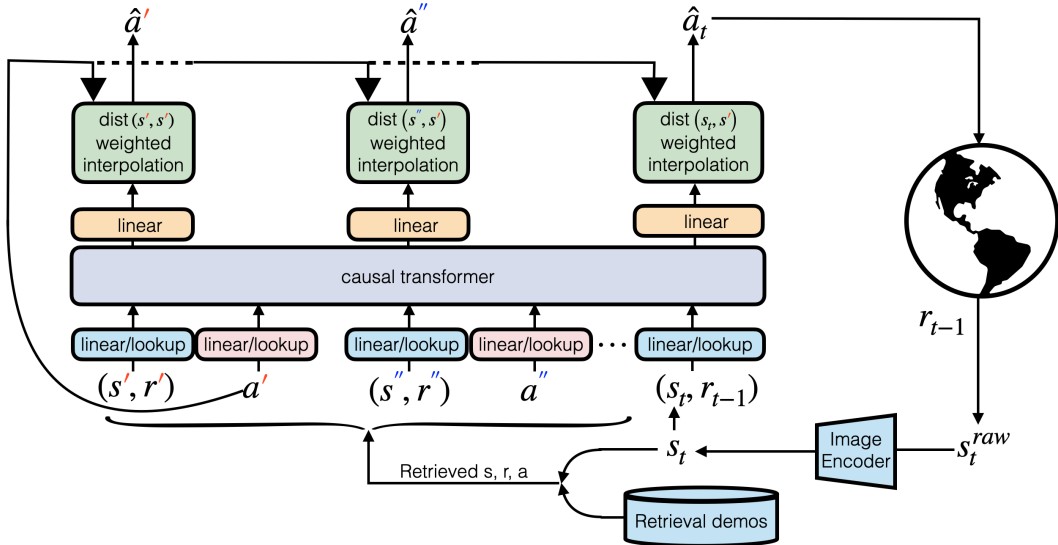

Figure 3: **The REGENT architecture and overview.** (1) A query state (from the unseen environment during deployment or from training environments' datasets during pre-training) is processed for retrieval. (2) The $n$ nearest states from a few demonstrations in an unseen environment or from a designated retrieval subset of pre-training environments' datasets are retrieved. These states, and their corresponding previous rewards and actions, are added to the context in order of their closeness to the query state, followed by the query state and previous reward. (3) The predictions from the REGENT transformer are combined with the first retrieved action. (4) At deployment, only the predicted query action is used. During pre-training, the loss from predicting all actions is used to train the transformer.

as described in Appendix A. As shown in Figure 3, for predicting each action, either for a retrieved state in the context or for the query state, we set $d(s, s')$ in Equations (1) and (3) to the distance between that state $s$ and the first (closest) retrieved state in the context $s'$.

Further, the exponential weights in Equation (1) allows us to smoothly transition between R&P and the parametric policy $\pi_\theta$. When the state $s_t$ is close enough to first retrieved state in the context $c_t$, $\pi_{\text{REGENT}}$ simply plays the retrieved action. However, as it moves further away, policy $\pi_{\text{R\&P}}$ becomes a uniform distribution and the parametric policy $\pi_\theta$ takes more precedence. *We also hypothesize that this interpolation allows the transformer to more readily generalize to unseen environments, since it is given the easier task of predicting the residual to the R&P action rather than predicting the complete action.*

**Pre-training REGENT and Loss Function**: We train REGENT by minimizing the total cross-entropy loss on discrete actions and total mean-squared error on continuous actions for all $n + 1$ action predictions ($n$ in the context and 1 query). We also follow the JAT recipe in ensuring that each training batch consists only of data from a single environment's dataset. We provide details about the training hyperparameters in Appendix A.

**REGENT Training Data and Environment Suites**: In the JAT/Gato setting, we pre-train on 100k transitions in each of the 45 Metaworld training environments, 9 Mujoco training environments, 52 Atari training environments, and 39 BabyAI training environments. This adds up to a total of 14.5M transitions used to pre-train REGENT. We obtain these transitions by taking a subset of the open-source JAT dataset [19]. The complete JAT dataset consists of 5-10x the amount of data we use in each environment. The Atari environments have discrete actions and four consecutive grayscale images as states. However, we embed each combined four-stacked image into a vector of 512 dimensions and use this image embedding everywhere (for R&P and REGENT). The details of embedding images are discussed later in this Section. The Metaworld and Mujoco environments have proprioceptive vector observations and continuous action spaces. The BabyAI environments have discrete observations, text specifying a goal in natural language, and discrete actions. We emphasize that the same generalist policy handles completely different observation and action modalities and dimensions across environments. The dimensions of observations & actions are given in Appendix A.

In the ProcGen setting that borrows from MTT [37], we generate 20 rollouts with PPO policies on each of the 12 training environments for various levels depending on the environment. We do so to ensure that the total number of transitions in each environment is 1M. The total size of the pre-training

dataset is 12M. All environments have an image-based state space and discrete actions. The number of levels varies from 63 at the smallest in bigfish to 1565 at the largest in maze. This variation arises from the difference in rollout horizon in each environment. We note that, even the largest value of 1565 is an order-of-magnitude fewer than the 10k levels in each game used by MTT for pre-training.

**REGENT Architecture**: REGENT adapts the JAT architecture in the JAT/Gato setting. It consists of a causal transformer trunk. It has a shared linear encoder for image embeddings, vector observations, and continuous actions. This encoder takes a maximum input vector of length 513, accommodating the largest input vectors—image embeddings of size 512—plus a single reward value. When the length of an input vector is less than that, it is cyclically repeated and padded to the maximum length. It has a large shared lookup table with the GPT2 [35] vocabulary size for encoding discrete actions, discrete observations, and text observations. Another linear layer is used to combine multiple discrete values and text tokens present in a single observation vector. It has a shared linear head for predicting continuous actions. A shared linear head is used for predicting distributions over discrete actions. All of the above components are shared across environments with but only a subset of input encoders and output heads may be triggered during a forward and backward pass depending on the input modalities in an environment and output action space. We note that states and rewards are concatenated together before encoding following the JAT recipe [19]. REGENT has a total of 138.6M parameters, including the frozen ResNet18 image encoder, compared to JAT's 192.7M parameters.

We simplify REGENT for the ProcGen setting keeping the transformer trunk with only a convolution encoder for direct image inputs; a lookup table for encoding discrete actions; and a linear head for predicting distributions over discrete actions. Here, following the MTT recipe, we do not use any rewards. We only use the states and actions. Here, REGENT has a total of 116M parameters compared to MTT's 310M parameters. We detail all architectural hyperparameters and the differences between JAT, MTT, and REGENT in Appendix A.

**Processing Raw Observations, Distance Metrics, Retrieval Mechanism, and Preprocessing Training Data**: Recall that we are evaluating in two settings, depicted in Figures 1 and 2. In the JAT/Gato setting, if the raw observations consist of images, we embed the image with an off-the-shelf ResNet18 encoder [13] trained only on ImageNet data. If instead, the raw observations are proprioceptive vectors, we use them directly without modification. In the ProcGen setting, the raw observations consist of images that we use directly without modification.

In the JAT/Gato setting, we use the $\ell_2$ distance metric to compute distances between pairs of observations, regardless of whether they are image embeddings or proprioceptive vectors. To significantly speed up the search for nearest states to a query state, we leverage similarity search indices [15]. In the ProcGen setting, we utilize the SSIM distance metric [51] to obtain distances between two images. We perform a naive search for the nearest images to a query image, however all SSIM distance calculations are parallelized on GPUs to speed up the retrieval process.

R&P simply performs the retrieval process described above at evaluation time, obtains the closest state to a query state, and plays the corresponding action. REGENT on the other hand has to setup its pre-training dataset as follows. It first designates a certain number of randomly chosen demonstrations per environment as the retrieval set in that environment. This is described in Appendix A. Then, for each state in each environment's training dataset, we retrieve the $n = 19$ closest states from the designated retrieval subset for that dataset. We ensure that none of the retrieved states are from the same demonstration as the query state. In this process, we convert our dataset of transitions to a dataset of (context, query state, query reward) datapoints where the context consists of 19 retrieved (state, reward, action) tuples. Now, we can begin pre-training REGENT on this dataset.

**Evaluating REGENT**: After pre-training, in each of the two settings, we have one model that can be deployed directly, without finetuning, on unseen environments. During deployment in an environment, at each step, we retrieve the $n = 19$ closest states to the current query state (and their corresponding actions and rewards). We pass this context with the query state, and previous reward into Equation (1) through the architecture described above to predict the query action to take in the environment.

In the JAT/Gato setting, for the unseen environments, we hold-out 5 Metaworld, 5 Atari, and two Mujoco environments (see Figure 1). In the ProcGen setting, following MTT [37], we hold-out 5 environments. Unlike MTT, we also evaluate on unseen levels in training environments (see Figure 2). We explain these choices in Appendix A. In this work, we focus on unseen environments in the same suite as training environments and leave further generalization to future work. Finally, we also

note that in all game-playing environments, we add a small sticky probability [31] of 0.05 in unseen Atari environments and 0.2 in unseen ProcGen environments following [37]. This is not present in any data or demonstration, which induces further stochasticity and tests the ability of both R&P and REGENT to truly generalize under novel and stochastic dynamics.

## 5 SUB-OPTIMALITY BOUND FOR REGENT POLICIES

In this section, we aim to bound the sub-optimality of the REGENT policy. This is measured with respect to the expert policy $\pi_j^*$, that generated the retrieval demonstrations $\mathcal{D}_j$. We focus on the discrete action case here and leave the continuous action case for future work. The sub-optimality gap in (training or unseen) environment $j$ is given by $(J(\pi_j^*) - J(\pi_{\text{REGENT}}^\theta))$. Inspired by the theoretical analysis of Sridhar et al. [46], we define the "most isolated state" and use this definition to bound the total variation in the REGENT policy class and hence the sub-optimality gap.

That is, first, given $\mathcal{D}_j$, we wish to obtain the maximum value of the distance term $d(s_t, s')$ in Equations (1) and (3). To do so, we define the most isolated state as follows.

**Definition 5.1** (Most Isolated State). For a given set of retrieval demonstrations $\mathcal{D}_j$ in environment $j$, we define the most isolated state $s_{\mathcal{D}_j}^I := \arg\max_{s \in S_j} \left( \min_{s' \in \mathcal{D}_j} d(s, s') \right)$, and consequently the distance to the most isolated state as $d_{\mathcal{D}_j}^I = \min_{s' \in \mathcal{D}_j} d(s_{\mathcal{D}_j}^I, s')$.

All distances between a state in this environment and its closest retrieved state are less than the above value, which also measures state space coverage by the demonstrations available for retrieval. Using the above definition, we have the following.

**Theorem 5.2.** *The sub-optimality gap in environment $j$ is*
$$J(\pi_j^*) - J(\pi_{\text{REGENT}}^\theta) \leq min\{H, H^2(1 - e^{-\lambda d_{\mathcal{D}_j}^I})\}$$

*Proof*: We provide the proof in Appendix B.

The main consequence of this theorem, also observed in our results, is that the sub-optimality gap reduces with more demonstrations in $\mathcal{D}_j$ because of the reduced distance to the most isolated state.

## 6 EXPERIMENTAL EVALUATION

In our experiments, we aim to answer the following key questions in the two settings depicted in Figures 1 and 2. (1) How well can R&P and REGENT generalize to unseen environments? (2) How does finetuning in the new environments improve REGENT? (3) How well can REGENT generalize to variations of the training environments and perform in aggregate on training environments? (4) How does REGENT qualitatively compare with R&P in using the retrieved context?

▶ **Metrics**: We plot the normalized return computed using the return of a random and expert agent in each environment as $\frac{\text{(total return−random return)}}{\text{(expert return−random return)}}$. The expert and random returns for all JAT/Gato environments are made available in the original work [19]. The expert and random returns for the ProcGen environments can be found in the original ProcGen paper [11].

▶ **Baselines**: We consider a different set of baselines for each of the two settings. In the JAT/Gato setting, we compare R&P and REGENT with two variants of JAT. The first is a JAT model trained on the same dataset as REGENT. The second is a JAT model trained on all available JAT data which consists of an order of magnitude more datapoints (in particular, 5x more data in Atari environments and 10x more data in all other environments). We label the former apples-to-apples comparison as JAT/Gato and the latter as JAT/Gato (All Data). We also compare with JAT/Gato with RAG at inference time. We use the same retrieval mechanism as REGENT for this baseline. Additional details and all hyperparameters for the baselines are in Appendix A.

In the ProcGen setting, we compare with MTT [37]. MTT only reports results on unseen environments and not on unseen levels in training environments. We simply take the best MTT result on each of the four unseen environments in [37], obtained when 4 demonstrations are included in the MTT context.

▶ **Finetuning and Train-from-scratch Baselines**: We fully finetune and parameter-efficient finetune (PEFT with IA3 [30]) both JAT/Gato and JAT/Gato (All Data) on various number of demonstrations in each unseen environment in the JAT/Gato setting and compare with REGENT. We also

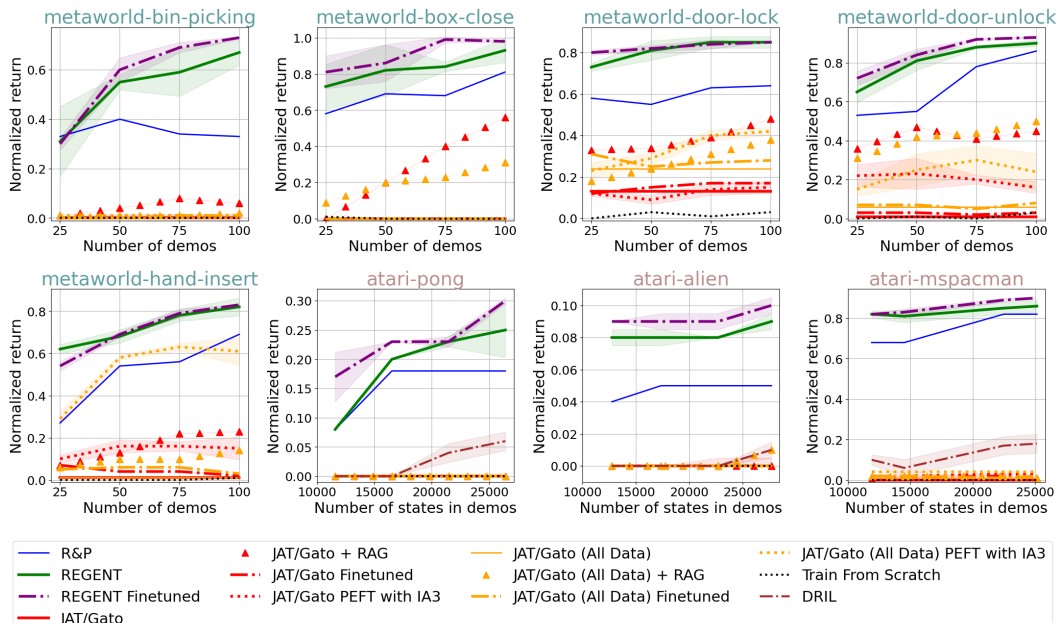

Figure 4: Normalized returns in the unseen Metaworld and Atari environments against the number of demonstration trajectories the agent can retrieve from or finetune on. Each agent is evaluated across 100 rollouts of different seeds in Metaworld and 15 rollouts of different seeds (with $p_{sticky} = 0.05$) in Atari. We compute the overall mean and standard deviation over three training seeds for REGENT, REGENT Finetuned, the PEFT with IA3 baselines, and DRIL. See Table 1 for detailed results.

compare with a train-from-scratch behavior cloning baseline and another imitation learning baseline (DRIL [6]). For completeness, we also finetune REGENT on various number of demonstrations in each unseen environment in the JAT/Gato setting. We use the same hyperparameters in each finetuning method across JAT/Gato variants and REGENT. Additional details are in Appendix A.

**Generalization to Unseen Environments**: We plot the normalized return obtained by all methods in unseen Metaworld and Atari environments from the JAT/Gato setting for various number of demonstrations (25, 50, 75, 100) in Figure 4. For a fair comparison across Atari environments with marked differences in episode horizons, we plot normalized return against various number of states in the demonstrations (atleast 10k, 15k, 20k, 25k) rather than the number of demonstrations itself. These demonstrations can be used by the different methods for retrieval or fine-tuning. We also plot the normalized return obtained by all methods in unseen ProcGen environments against various number of demonstrations (2, 4, 8, 12, 16, 20) to retrieve from in Figure 5. As mentioned before in Section 4.2, unseen Atari environments have a sticky probability of 0.05 and unseen ProcGen environments have a sticky probability of 0.2.

In both Figures 4 and 5, we observe that R&P and REGENT can generalize well to unseen Atari and ProcGen environments with image observations and discrete actions as well as to unseen Metaworld environments with vector observations and continuous actions. R&P is a surprisingly strong baseline, but REGENT improves on R&P consistently. In general, both methods appear to steadily improve with more demonstrations (with only a few exceptions).

In Figure 4, we observe that JAT/Gato cannot generalize to most unseen environments. REGENT (and even R&P) outperform even the All Data variants of JAT/Gato which were pre-trained on 5-10x the size of the REGENT dataset. The sticky probability in unseen Atari environments adds further stochasticity and stress tests true generalization against simply replaying demonstrations. The JAT/Gato + RAG baselines actually improve over zero-shot JAT/Gato but are still significantly outperformed by REGENT. This demonstrates the advantage of REGENT's architecture and retrieval augmented pretraining over just performing RAG at inference time.

In Figure 5, REGENT (and even R&P) outperform MTT which was trained on an order of magnitude more training levels and data than REGENT while also having approximately 3x the parameters.

We also note that without the interpolation between R&P and the output of the transformer, RE-GENT does not generalize to unseen environments (i.e., it performs like a random agent on unseen environments). This interpolation is key to REGENT.

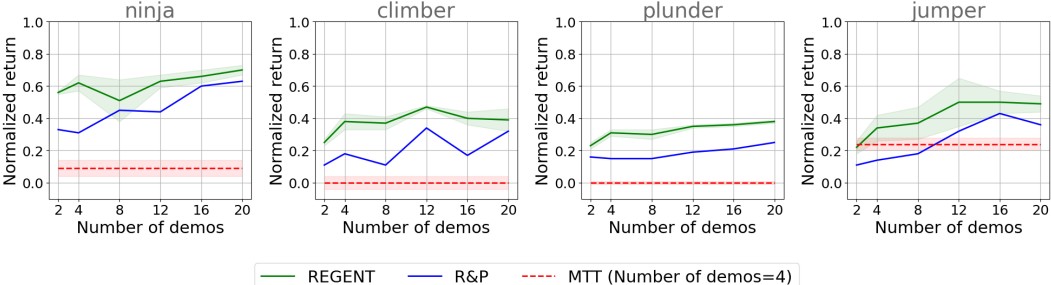

Figure 5: Normalized returns in unseen ProcGen environments against the number of demonstration trajectories the agent can retrieve from. REGENT and R&P agents are evaluated across 10 levels with 5 rollouts and $p_{\text{sticky}} = 0.2$. We compute the overall mean and standard deviation over three training seeds for REGENT. The values for MTT are the best scores reported in [37]. See Table 7 for detailed results.

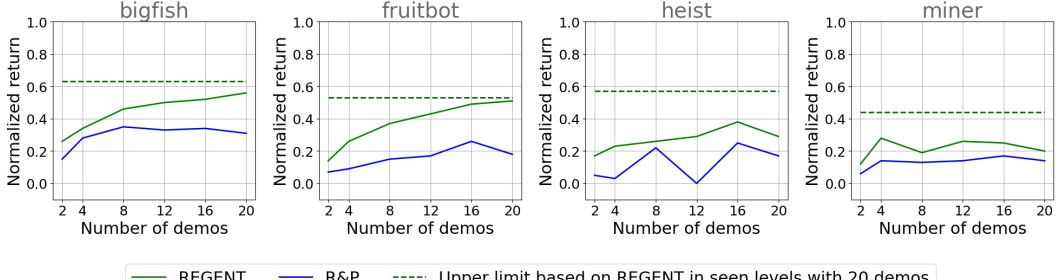

Figure 6: Normalized returns in unseen levels in 4 (of 12) ProcGen training environments against the number of demonstration trajectories the agent can retrieve from. Each value is an average across 10 levels with 5 rollouts each with $p_{\text{sticky}} = 0.1$. The other 8 plots are similar to the above and can be found in Figure 19 (in Appendix D). See Table 8 for detailed results.

**Effect of Finetuning**: From Figure 4, we see that JAT/Gato, along with its All Data variant, struggles to perform in all unseen environments even after full finetuning. JAT/Gato performs slightly better after PEFT with IA3 in the unseen metaworld environments. Training from scratch on the few demonstrations in each unseen environment also does not obtain any meaningful performance in most environments. REGENT (and even R&P), without any finetuning, outperforms all finetuned variants of JAT/Gato. Moreover, we can see that REGENT further improves after finetuning, even with only a few demonstrations.

To highlight REGENT's ability to adapt from very little data, we note that in Figure 4, we only vary the number of states in Atari demonstrations until 25k. Whereas, the closest generalist policy that finetunes to new Atari environments, the multi-game decision transformer [28], requires 1M transitions.

Finally, we note that all methods face a hurdle in generalizing to the very long-horizon atari-spaceinvaders and atari-stargunner environments, which have horizons about 10x that of atari-pong and hence have not been shown in Figure 4. Generalization to the new embodiments in the two unseen Mujoco environments proves equally challenging, we discuss this in detail in Appendix C.

**Generalization to Variations of the Training Environments**: In the ProcGen setting in Figure 2, the unseen levels of the 12 training environments provide an avenue to test a middle ground of generalization between training and unseen environments. We plot the normalized returns on unseen levels of 4 (of 12) training environments (with a sticky probability of 0.1) in Figure 6. The remaining can be found in Figure 19 (Appendix D) and are similar to the above 4. In Figure 6, we again observe that REGENT performs the best in unseen levels while R&P remains a strong baseline. We also depict the performance of REGENT on seen levels of these training environments with a dotted line in this Figure. This represents an upper bound for REGENT's performance in unseen levels. We observe that with a large number of demonstrations, REGENT appears to reach close to this upper bound simply via retrieval-augmentation and in-context learning in some environments.

**Aggregate Performance on Training Environments**: We plot the aggregate normalized return, both IQM [1] and mean, on training environments for each of the 4 suites (in the JAT/Gato setting) in Figure 7. We notice that REGENT significantly exceeds JAT/Gato on Metaworld, exceeds it on Atari, matches it on Mujoco, and is close to matching it on BabyAI. This demonstrates the dual advantage of REGENT which gains the capability to generalize to unseen environments while preserving overall multi-task performance in training environments.

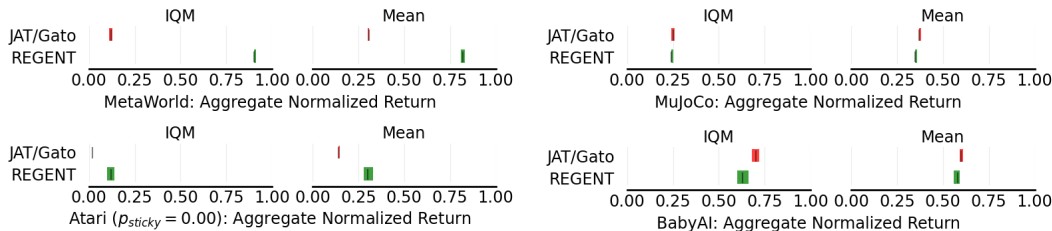

Figure 7: Aggregate normalized returns in the 45 training Metaworld environments, 9 training Mujoco environments, 52 training Atari environments, and 39 training BabyAI environments. See Figures 15, 16, 17, 18 (in Appendix C) for performance on each training environment of each suite.

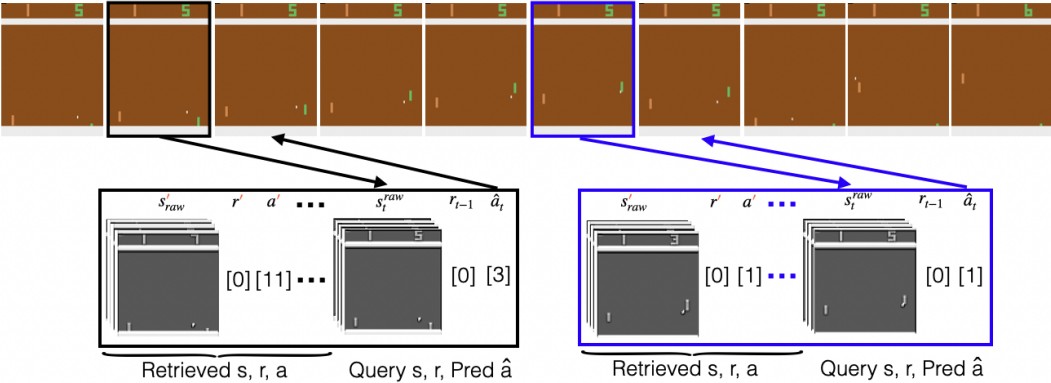

Figure 8: Qualitative examples of a few inputs and outputs of REGENT for two states in a rendered rollout in the unseen, discrete action space, atari-pong environment. REGENT leverages its in-context learning capabilities and interpolation with R&P to either make a simple decision and predict the same action as R&P (see blue box on the right) or predict better actions at key states (see black box on the left) that leads to better overall performance as seen in Figure 4.

**Qualitative Examples**: We plot examples of the inputs and outputs of REGENT at various states during a rendered rollout in the atari-pong environment in Figure 8 to highlight REGENT's learned in-context learning capabilities and interpolation with R&P. We also provide a qualitative example in a continuous metaworld environment in Appendix C.

**Ablations and Discussions**: We performed ablations on REGENT's context length, context order, and distance metric in Figures 9, 10, and 12 in Appendix C. We found that a larger context length improves performance (but saturates quickly), the order of states in the context is key especially when unseen environments significantly differ from pretraining environments (such as in Atari), and that $\ell_2$ outperforms cosine distance. We also discuss inference runtimes in Figure 11 in Appendix C.

## 7 CONCLUSIONS AND FUTURE WORK

In this work, we demonstrated that retrieval offers generalist agents a powerful bias for rapid adaptation to new environments, even without large models and vast datasets. We showed that a simple retrieval-based 1-nearest neighbor agent, Retrieve and Play (R&P), is a strong baseline that matches or exceeds the performance of today's state-of-the-art generalist agents in unseen environments. Building on this, we proposed a semi-parametric generalist agent, REGENT, that pre-trains a transformer-based policy on sequences of query state, reward, and retrieved context. REGENT exploits retrieval-augmentation and in-context learning for direct deployment in unseen environments with only a few demonstrations. Even after pre-training on an order of magnitude fewer datapoints than other generalist agents and with fewer parameters, REGENT outperforms them across unseen environments and surpasses them even after they have been finetuned on demonstrations from those unseen environments. REGENT, itself, further improves with finetuning on even a small number of demonstrations. We note that REGENT faces a couple of limitations: adapting to new embodiments and long-horizon environments. In future work, we believe a larger diversity of embodiments in the training dataset and improved retrieval from longer demonstrations can help REGENT overcome these challenges. We conclude with the conviction that retrieval in general and REGENT in particular redefines the possibilities for developing highly adaptive and efficient generalist agents, even with limited resources.

**Acknowledgements:** This work was supported in part by ARO MURI W911NF-20-1-0080, NSF 2143274, NSF CAREER 2239301, NSF 2331783, and DARPA TIAMAT HR00112490421. Any opinions, findings, conclusions or recommendations expressed in this material are those of the authors and do not necessarily reflect the views the Army Research Office (ARO), the Department of Defense, or the United States Government.

**Reproducibility statement:** To ensure reproducibility, we have released all code at `https://kaustubhsridhar.github.io/regent-research`. We have also described all datasets, environment suites, baselines, and REGENT itself in detail in Section 4, Section 6, and Appendix A.

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

APPENDIX

## A  ADDITIONAL DETAILS FOR REGENT AND VARIOUS BASELINES

### Additional Details for REGENT

**MixedReLU activation**: MixedReLU is a tanh-like activation function from [46] given by $\text{MixedReLU}(x) = (2(\text{ReLU}(\frac{x+1}{2}) - \text{ReLU}(\frac{x-1}{2})) - 1)$ which simplifies to $-1$ for $x < -1$, $x$ for $-1 \leq x \leq 1$, and $1$ for $x > 1$.

**Normalizing distances to** $[0, 1]$: We compute the 95th percentile of all distances $d(s, s')$ between any (retrieved or query) state $s$ and the first (closest) retrieved state $s'$. This value is computed from the demonstrations $\mathcal{D}_j$ and is used to normalize all distances in that environment. This value is calculated for all (training or unseen) environments during the preprocessing stage. If after normalization, a distance value is greater than 1, we simply clip it to 1.

**Softening $\pi_{\text{R\&P}}$ logits for discrete action environments**: Let us assume that the discrete action space consists of $N_{\text{act}}$ different actions. We have the following.

$$\pi_{\text{R\&P}}(a|s_t, c_t) = \begin{cases} \frac{1+(N_{\text{act}}-1)(1-d(s_t, s'))}{N_{\text{act}}} & \text{, if } a = a' \\ \frac{d(s_t, s')}{N_{\text{act}}} & \text{, if } a \neq a' \end{cases} \tag{3}$$

The distribution induced by the modified $\pi_{\text{R\&P}}$ function assigns all probability mass to the action $a'$ when $d(s_t, s') = 0$ and acts like a uniform distribution when $d(s_t, s') = 1$, thereby preventing bias toward any single logit.

**Architectural hyperparameters**: In both settings, the transformer trunk consists of 12 layers and 12 heads with a hidden size of 768. We set the maximum position encodings to 40 for 20 (state, previous reward)'s and 20 actions. Of these 20, 19 belong to the retrieved context and 1 belongs to the query. In the JAT/Gato setting, the maximum multi-discrete observation size is 212 (for BabyAI). The maximum continuous observation size as discussed before in Section 4 is set to 513, a consequence of the ResNet18 image embedding models's [13] embedding size of 512 with an additional 1 dimension for the reward. All linear encoding layers map from their corresponding input size to the hidden size. Following the JAT model, the linear decoder head for predicting continuous actions maps from the hidden size to the maximum continuous size discussed above (513). On the other hand, the linear decoder head for predicting distributions over discrete actions maps from the hidden size to only the maximum number of discrete actions across all discrete environments (18), whereas in the JAT model, they map to the full GPT2 vocab size of 50257.

In the JAT/Gato setting, the original JAT architecture has a much larger context size (that is not required for REGENT), a larger image encoder (than the resnet18 used in REGENT), a much larger discrete decoder predicting distributions over the entire GPT2 vocab size (instead of just the maximum number of discrete actions in REGENT), and has (optional) decoders for predicting various observations (that is also not required for REGENT).

In the ProcGen setting, MTT has a 18 layers, 16 heads, and a hidden size of 1024 – all three of which are much larger than REGENT's architectural hyperparameters mentioned above.

**Training hyperparameters**: In the JAT/Gato setting, we use a batch size of 512 and the AdamW optimizer with parameters $\beta_1 = 0.9$ and $\beta_2 = 0.999$. The learning rate starts at 5e-5 and decays to zero through 3 epochs over all pre-training data. We early stop after a single epoch because we find that overtraining reduces in-context learning performance. This observation is consistent with similar observations about the transience of in-context learning in literature [42].

In the ProcGen setting, we use a batch size of 1024, a starting learning of 1e-4 also decaying over 3 epochs over all the pre-training data with an early stop after the first epoch. We again use the AdamW optimizer with parameters $\beta_1 = 0.9$ and $\beta_2 = 0.95$.

**Designating retrieval demonstrations in each training environment's dataset**: In the JAT/Gato setting, we designate 100 random demonstrations from the each of training vector observation environments as the retrieval subset for that environment. We also designate 5 random demonstrations from the training image environments as the retrieval subsets for said environments. In the ProcGen

setting, with only 20 demonstrations per level per environment, we designate all 20 as the retrieval set for that level in that environment.

We use all states in all demonstrations in each training environment's dataset as query states. So, when a query state is from a demonstration in the designated retrieval subset, we simply retrieve from all other demonstrations in the designated retrieval subset except the current one. When a query state is from a demonstration not in the designated retrieval subset, we retrieve simply from the designated retrieval subset. This way, we ensure that none of the retrieved states are from the same demonstration as the query state. A possible direction for future work includes intelligently designating and collecting demonstrations for retrieval (perhaps using ideas like persistency of excitation [45], memory-based methods [56, 46, 47, 34], expert intervention methods [16]).

**Choice of held-out environments**: In the `JAT`/`Gato` setting, for the unseen environments, we hold-out the 5 Metaworld environments recommended in [58], the 5 Atari environments held-out in [28], and finally, we choose two Mujoco environments of mixed difficulty (see Figure 1). In the ProcGen setting, following MTT [37], we hold-out 5 environments.

In this work, we generalize to unseen environments from the same suites as the training environments. In future work, it would be interesting to examine what is needed for generalization to new suites (such as more manipulation suites [62], quadrotor, driving simulators [12, 14, 44], biological simulation [32], etc.).

**Additional details on all environments:** In the `JAT`/`Gato` setting, the Atari environments have 18 discrete actions and four consecutive grayscale images as states. However, we embed each combined four-stacked image, with an off-the-shelf ResNet18 encoder [13], into a vector of 512 dimensions and use this image embedding everywhere (for `R&P` and `REGENT`). All Metaworld environments have observations with 39 dimensions and actions with 4 dimensions. Mujoco environments have observations with 4 to 376 dimensions and actions with 1 to 17 dimensions. The BabyAI environments have up to 7 discrete actions. They have discrete observations and text observations. After tokenizing the BabyAI text observations, which specify a goal in natural language, into 64 tokens, BabyAI observations have 212 total dimensions and consist only of discrete values.

### Additional Details on Various Baselines

**Additional information on re-training `JAT`/`Gato`**: When retraining `JAT`/`Gato`, we follow all hypepermaters chosen in [19]. We note that we skip the VQA datasets in our retraining of `JAT`/`Gato`, and use a smaller batch size to fit available GPUs. For `JAT`/`Gato` (All Data) we train over the full `JAT` dataset for the specified 250k training steps. For `JAT`/`Gato` we train on the `REGENT` subset (which is 5-10x smaller) for 25k training steps. In both variants, the training loss converges to the same low value at about 5000 steps into the training run.

**Additional information on `JAT`/`Gato` + RAG (at inference):** We note that `JAT`/`Gato` is like a typical LLM— it has a causal transformer architecture and undergoes vanilla pretraining on consecutive states-rewards-actions. In both the `JAT`/`Gato` + RAG baselines, we use the same retrieval mechanism as REGENT. We provide the retrieved states-rewards-actions context to these RAG baselines replacing the context of the past few states-rewards-actions in vanilla `JAT`/`Gato` baselines.

REGENT significantly outperforms both `JAT`/`Gato` + RAG baselines in Figure 4. This demonstrates the advantage of REGENT's architecture and retrieval augmented pretraining over just performing RAG at inference time. But, the `JAT`/`Gato` + RAG baselines outperform vanilla `JAT`/`Gato` as well as the `JAT`/`Gato` fully Finetuned baselines in the unseen metaworld environments. From this we can conclude that RAG at inference time is also important for performance.

**Additional information on `JAT`/`Gato` Finetuned and `REGENT` Finetuned**: Starting from a pre-trained checkpoint, we finetune the full `REGENT` and `JAT`/`Gato` checkpoints using the same optimizer as pre-training time but starting with 1/10th the learning rate (*i.e.,* 5e-6) and for 3 epochs over the finetuning demonstrations. We use this same recipe across all environments and for both `REGENT` and `JAT`/`Gato`. We only use 3 epochs to prevent the overwriting of any capabilities learned during pre-training.

**Additional information on Parameter Efficient Finetuing (PEFT) of `JAT`/`Gato` with IA3 [30]:** We finetuned `JAT`/`Gato` and `JAT`/`Gato` (All Data) with the IA3 PEFT method [30] and plotted the

results in Figure 4. We used the huggingface implementation and the hyperparameters given in the IA3 paper (i.e. 1000 training steps, batch size of 8, and the adafactor optimizer with 3e-3 learning rate and 60 warmup steps in a linear decay schedule). We found that finetuning `JAT/Gato` with IA3 improved performance over our simple full finetuning recipe, especially in 3 metaworld environments. REGENT still outperforms both the IA3 finetuned `JAT/Gato` and the fully finetuned `JAT/Gato` in all unseen environments in Figure 4.

**Additional information on the train-from-scratch behavior cloning baseline**: For the policy in this baseline, we use the Impala CNN [18] in Atari environments and a MLP with two hidden layers with 256 neurons each in Metaworld and Mujoco environments. We use the same learning rate (5e-5) for the same 3 epochs with the same optimizer (AdamW with $\beta_1 = 0.9$ and $\beta_2 = 0.999$) as used for REGENT except we use a constant learning rate schedule.

**Additional information on DRIL:** We added an imitation learning baseline, DRIL [6], to the atari environments in Figure 4. Please note that DRIL consists of two stages. The first stage is pre training with a BC loss. In the second stage, the DRIL ensemble policy is allowed to interact in the online environment where it reduces the variance between the members of its ensemble with an online RL algorithm. We run the second online stage for 1M steps or around 1000 episodes. REGENT and all the other methods in Figure 4, on the other hand, are completely offline and never interact with any environment for training. While DRIL improves over the BC baseline, REGENT still outperforms DRIL in all three unseen atari environments.

**Additional information on MTT:** The MTT model weights are closed-source. In their ICML publication, the authors do not provide results on new levels. MTT is also difficult to train from scratch in any setting since they put multiple full trajectories in the context of the transformer, leading to exploding GPU VRAM requirements beyond academic compute resources. However, we simply report the best values from their paper in their exact setting to demonstrate REGENT's significantly improved performance even with a 3x smaller model and an order-of-magnitude less pretraining data. Similarly, we compare with `JAT/Gato` in their exact setting and significantly outperform `JAT/Gato` and `JAT/Gato` Finetuned, even though `JAT/Gato` has 1.5x our parameters and 5-10x our data.

We note that it is likely that MTT performance is close to random on a couple of unseen ProcGen environments only because the authors are forced to constrain MTT rollouts to 200 steps, stopping short of the true horizon of these environments. This is simply because of the quadratically increasing complexity of attention with increase in context size. MTT also likely does not report results beyond 4 demonstrations in the context for this reason. Yet, this is not an issue for `R&P` or REGENT which can scale to any number of demonstrations to retrieve from since it only uses up to the 19 closest states in the context.

## B  PROOF OF THEOREM 5.2

To prove theorem 5.2, we first state and prove the following lemma.

Let us refer to the family of policies represented by $\pi^\theta_{\text{REGENT}}(a|s,r,c)$ in Equation (1) for various $\theta$ as $\Pi^\theta(a|s,r,c)$. The parameters $\theta$ of the transformer are drawn from the space $\Theta$.

**Lemma B.1.** *(Total Variation in the Policy Class) The total variation of the policy class $\Pi^\theta(a|s,r,c)$ in environment $j$, $\forall\ \theta_1,\theta_2\ \in\ \Theta$  and for some $s\ \in\ \mathcal{S}_j,\ \ r\ \in\ \mathcal{R}_j,\ \ c\ \in\ \mathcal{C}_j,\ \ $ defined as $\sup_{\theta_1,\theta_2}\ \sup_a |\pi^{\theta_1}_{\text{REGENT}}(a|s,r,c) - \pi^{\theta_2}_{\text{REGENT}}(a|s,r,c)|$, is upper bounded by $(1 - e^{-\lambda d^I_{\mathcal{D}_j}})$.*

*Proof*: We have,

$$\sup_{\theta_1,\theta_2}\ \sup_a |\pi^{\theta_1}_{\text{REGENT}}(a|s,r,c) - \pi^{\theta_2}_{\text{REGENT}}(a|s,r,c)|$$

$$= \sup_{\theta_1,\theta_2}\ \sup_a (1 - e^{-\lambda d})\bigg(\mathsf{Softmax}(\pi_{\theta_1}(a|s,r,c)) - \mathsf{Softmax}(\pi_{\theta_2}(a|s,r,c))\bigg) \quad (4)$$

$$= (1 - e^{-\lambda d})\sup_{\theta_1,\theta_2}\ \sup_a \bigg(\mathsf{Softmax}(\pi_{\theta_1}(a|s,r,c)) - \mathsf{Softmax}(\pi_{\theta_2}(a|s,r,c))\bigg) \quad (5)$$

$$\leq (1 - e^{-\lambda d^I_{\mathcal{D}_j}}) \quad (6)$$

The first equality holds because for the same tuple of (state, previous reward, context), the `R&P` component does not change since it is not affected by the choice of parameters. The last inequality can be interpreted as a property of `Softmax` and the distance to the most isolated state. □

Now, we rewrite theorem 5.2 below and prove it.

**Theorem B.2.** *The sub-optimality gap in environment $j$ is*
$$\mathbf{J}(\pi_{\mathbf{j}}^*) - \mathbf{J}(\pi_{REGENT}^{\theta}) \leq \min\{\mathbf{H}, \mathbf{H^2}(\mathbf{1} - \mathbf{e}^{-\lambda \mathbf{d}_{\mathcal{D}_\mathbf{j}}^{\mathbf{I}}})\}$$

*Proof*: Recall that in imitation learning, if the population total variation (TV) risk $\mathbb{T}(\hat{\pi}, \pi^*) \leq \epsilon$, then, $J(\pi^*) - J(\hat{\pi}) \leq min\{H, H^2\epsilon\}$ (See [36] Lemma 4.3). Using this with Lemma B.1 proves the theorem. □

The main consequence of this theorem, also observed in our results, is that the sub-optimality gap reduces with more demonstrations in $\mathcal{D}_j$ because of the reduced distance to the most isolated state.

# C  ADDITIONAL JAT/GATO RESULTS

**Effect of number of states/rewards/actions in the context:** We plot the normalized return obtained by REGENT for different context sizes at evaluation time in unseen environments in Figure 9. We find that, in general, a larger context size leads to better performance. We also notice that the improvement in performance saturates quickly in many environments. Our choice of 20 states in the context (including the query state) leads to the best performance overall.

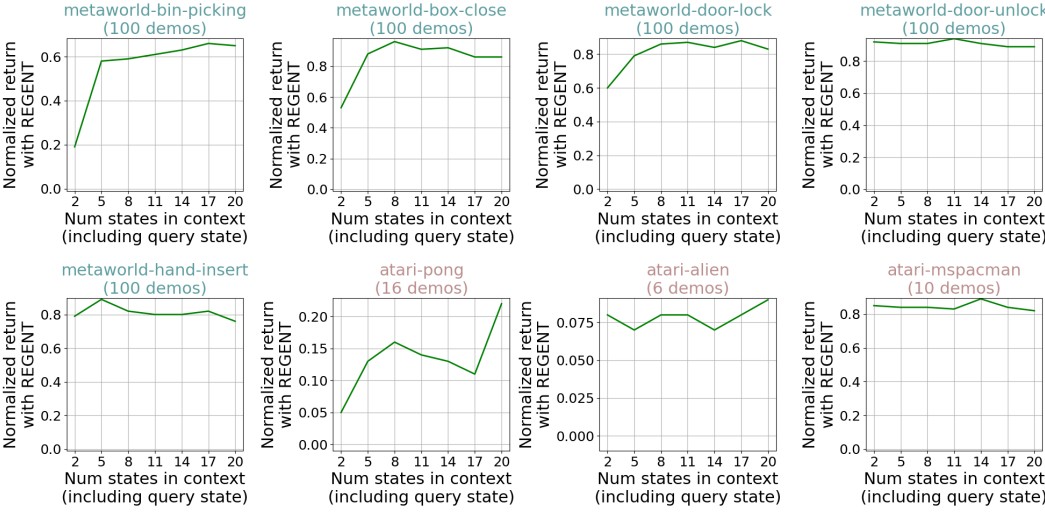

Figure 9: Normalized returns obtained by REGENT in the unseen Metaworld and Atari environments with a fixed retrieval buffer against the number of states given in-context (including the query state). Please note that the context size is $2n - 1$ if there are $n$ states in the context. Each value is from a single model training seed but evaluated over 100 rollouts of different seeds in Metaworld and 15 rollouts of different seeds (with $p_{\text{sticky}} = 0.05$) in Atari.

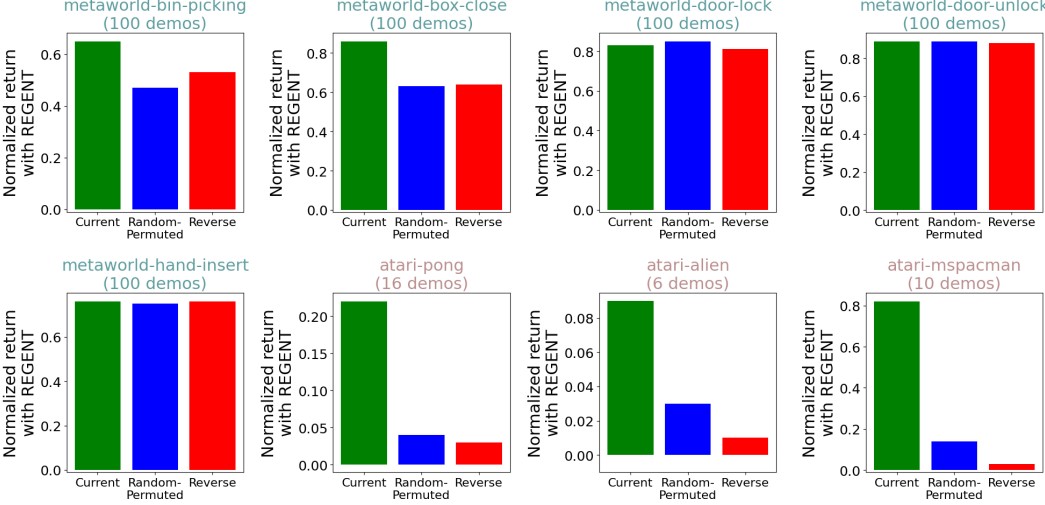

Figure 10: Normalized returns obtained by REGENT in the unseen Metaworld and Atari environments with a fixed retrieval buffer against the choice of ordering of states in the context. Each value is from a single model training seed but evaluated over 100 rollouts of different seeds in Metaworld and 15 rollouts of different seeds (with $p_{\text{sticky}} = 0.05$) in Atari.

**Effect of the ordering of the context:** We plot the normalized return obtained by REGENT for different choices of context ordering at evaluation time in unseen environments in Figure 10. We evaluated three options: "current", "random-permuted", and "reverse". In the "current" order, the retrieved tuples of (state, previous reward, action) are placed in order of their closeness to the query state $s_t$. In "random-permuted", the retrieved tuples are placed in a randomly chosen order in the context.

In "reverse", the "current" order of the retrieved tuples is reversed with the furthest state in the 19 retrieved states placed first. Our current choice performs the best overall. It significantly outperforms the other two orderings in the more challenging task of generalizing to unseen Atari games which have different image observations, dynamics, and rewards. In some metaworld environments on the other hand, all three orderings have similar performance denoting that all the 19 retrieved states for each query state are very similar to each other.

**Inference runtime values:** In Figure 11, we plot the inference runtime values of `REGENT`'s two main components in a metaworld environment: retrieval from demonstrations and forward propagation through the `REGENT` architecture. We find that retrieval takes between 2 to 8 milliseconds of time (for increasing number of demonstrations to retrieve from) while forward propagation takes around 8 milliseconds uniformly. Both these runtimes are small enough to allow real-time deployment of `REGENT` on real robots.

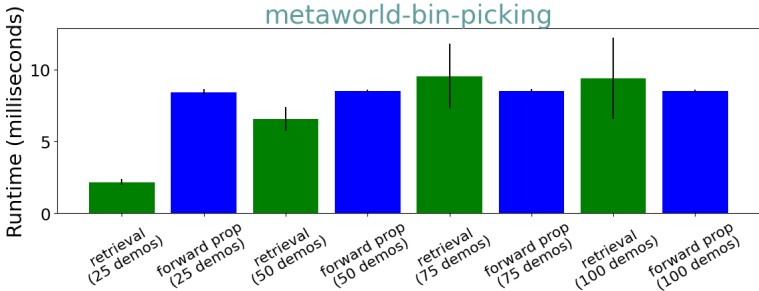

Figure 11: Average runtime values for retrieval and forward propagation through `REGENT` in metaworld for various number of demonstrations in the retrieval buffer.

**Effect of the distance metric:** We compared REGENT with cosine distance everywhere (from pretraining to evaluation) against our current version of REGENT with $\ell_2$ distance everywhere in the `JAT/Gato` setting in Figure 12. REGENT with cosine distance performs similarly to REGENT with $\ell_2$ distance in 3 unseen metaworld environments. In the other unseen metaworld environments as well as the more challenging unseen atari environments, REGENT with $\ell_2$ distance significantly outperforms REGENT with cosine distance. We speculate that this is because the magnitude of observations is important with both image embeddings and proprioceptive states and this is only taken into account by the $\ell_2$ distance (not the cosine distance).

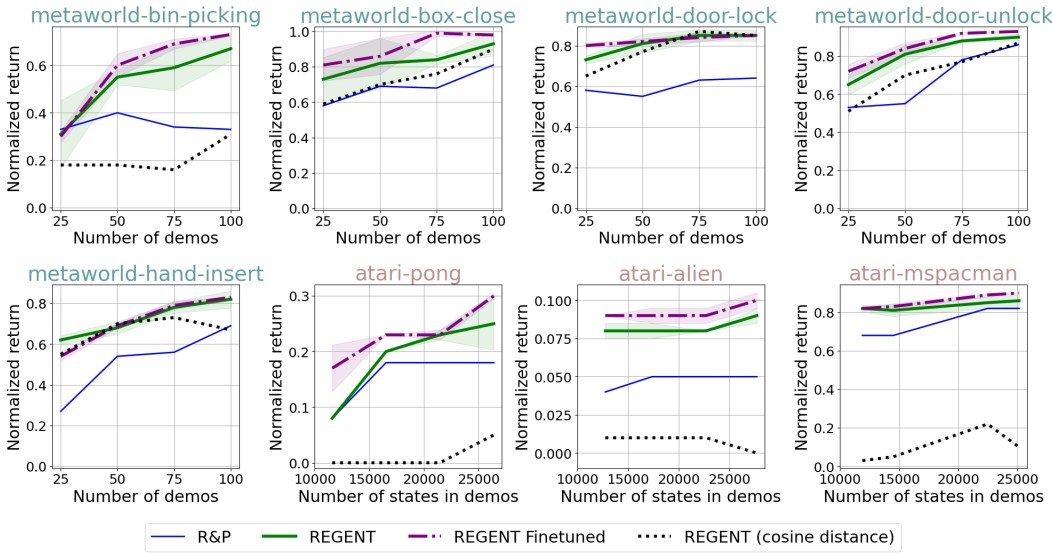

Figure 12: Normalized returns in the unseen Metaworld and Atari environments against the number of demonstration trajectories the agent can retrieve from. The 'REGENT (cosine distance)' values are an average over 100 rollouts of different seeds in Metaworld and 15 rollouts of different seeds (with $p_{\text{sticky}} = 0.05$) in Atari.

**Generalization to unseen Mujoco environments/embodiments:** In the two unseen Mujoco tasks shown in Figure 13, R&P appears to be a strong baseline even at generalizing to new embodiments! Moreover, after finetuning on just 25 demonstrations, only REGENT, not JAT/Gato, significantly improves to outperform other methods and only continues to get better with more demonstrations.

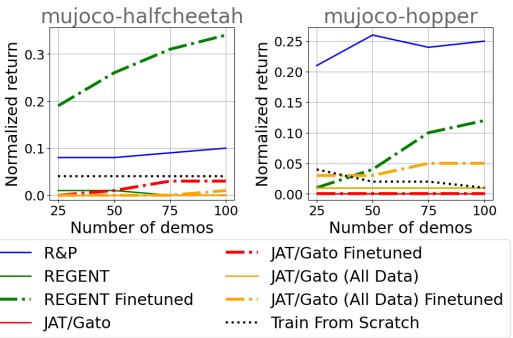

Figure 13: Normalized returns in the unseen Mujoco environments against the number of demonstration trajectories the agent can retrieve from or finetune on. Each value is an average across 100 rollouts of different seeds.

**Additional qualitative examples:** We plot examples of the inputs and outputs of REGENT at various states during a rendered rollout in the metaworld-bin-picking environment in Figure 14. In this continuous Metaworld example, REGENT can be seen predicting actions that are somewhat similar to but not the same as the first retrieved (R&P) action. These differences lead to better overall performance as seen in Figure 4. These differences are also the direct result of REGENT's in-context learning capabilities, learned from the preprocessed datasets from the pre-training environments.

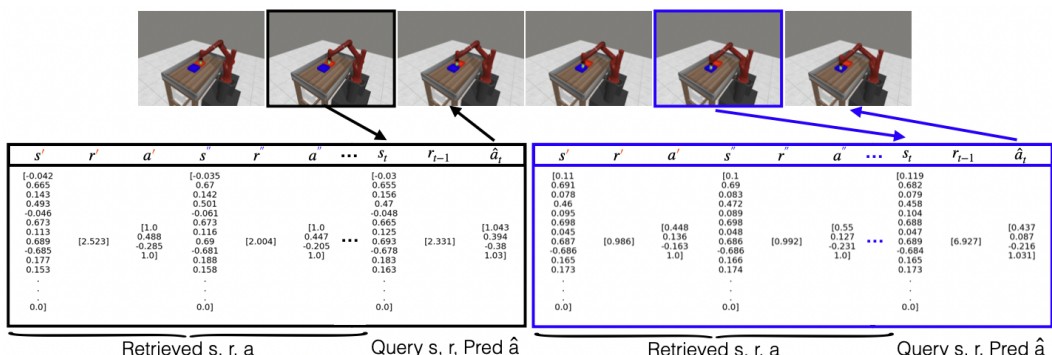

Figure 14: Qualitative examples of a few inputs and outputs of REGENT for various states in a rendered rollout in the unseen, continuous action space, metaworld-bin-picking environment.

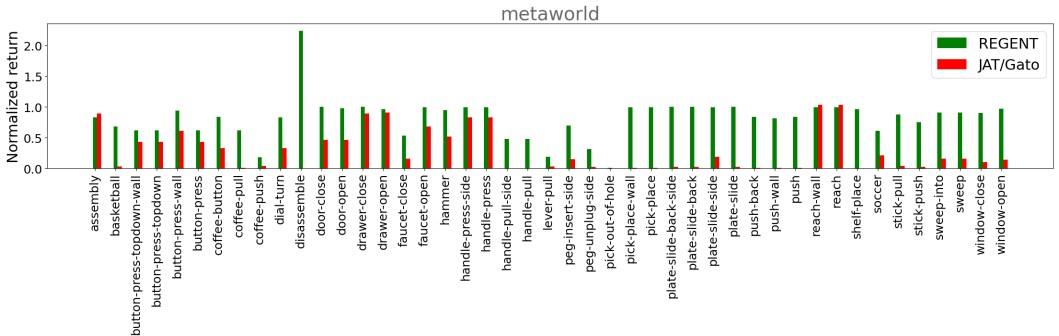

Figure 15: Normalized returns in each of the 45 seen Metaworld envs. Each value is an average across 100 rollouts of different seeds. See Table 3 for all values.

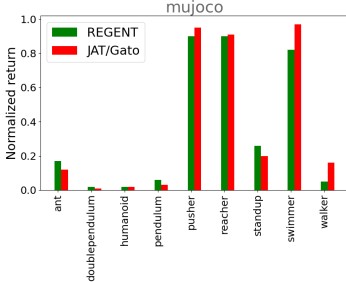

Figure 16: Normalized returns in each of the 9 seen Mujoco envs. Each value is an average across 100 rollouts of different seeds. See Table 4 for all values.

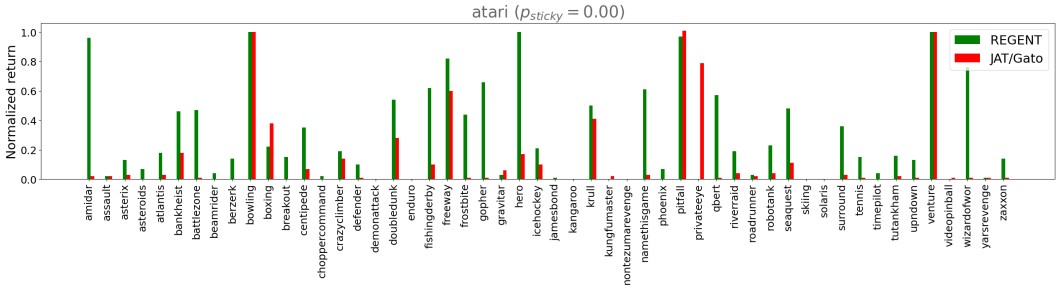

Figure 17: Normalized returns in each of the 52 seen Atari envs. Each value is an average across 15 rollouts of different seeds. See Table 5 for all values.

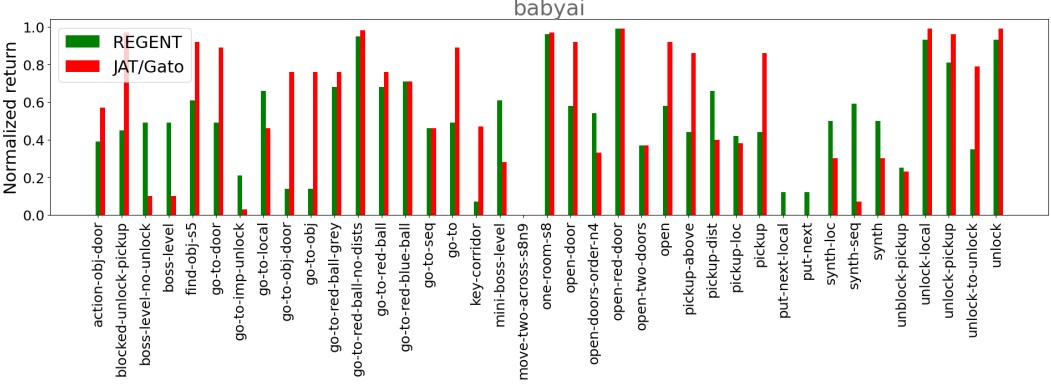

Figure 18: Normalized returns in each of the 45 seen BabyAI envs. Each value is an average across 100 rollouts of different seeds. See Table 6 for all values.

| Env | Num demos | R&P | REGENT | REGENT Finetuned | JAT/Gato | JAT/Gato + RAG | JAT/Gato Finetuned | JAT/Gato PEFT with IA3 | JAT/Gato (All Data) | JAT/Gato (All Data) + RAG | JAT/Gato (All Data) Finetuned | JAT/Gato (All Data) PEFT with IA3 | Train From Scratch | DRIL |
|---|---|---|---|---|---|---|---|---|---|---|---|---|---|---|
| metaworld-bin-picking | 25 | 0.33 | $0.31 \pm 0.142$ | $0.3 \pm 0.025$ | 0.0 | 0.01 | 0.0 | $0.0 \pm 0.005$ | 0.0 | 0.01 | 0.0 | $0.01 \pm 0.0$ | 0.0 | - |
| metaworld-bin-picking | 50 | 0.4 | $0.55 \pm 0.031$ | $0.6 \pm 0.049$ | 0.0 | 0.04 | 0.0 | $0.01 \pm 0.005$ | 0.0 | 0.01 | 0.0 | $0.01 \pm 0.0$ | 0.0 | - |
| metaworld-bin-picking | 75 | 0.34 | $0.59 \pm 0.097$ | $0.69 \pm 0.021$ | 0.0 | 0.08 | 0.01 | $0.01 \pm 0.005$ | 0.0 | 0.01 | 0.01 | $0.01 \pm 0.0$ | 0.0 | - |
| metaworld-bin-picking | 100 | 0.33 | $0.67 \pm 0.051$ | $0.73 \pm 0.005$ | 0.0 | 0.06 | 0.0 | $0.01 \pm 0.005$ | 0.0 | 0.02 | 0.01 | $0.01 \pm 0.0$ | 0.0 | - |
| metaworld-box-close | 25 | 0.58 | $0.73 \pm 0.123$ | $0.81 \pm 0.09$ | 0.0 | 0.0 | 0.0 | $0.0 \pm 0.0$ | 0.0 | 0.09 | 0.0 | $0.0 \pm 0.0$ | 0.01 | - |
| metaworld-box-close | 50 | 0.69 | $0.82 \pm 0.142$ | $0.86 \pm 0.102$ | 0.0 | 0.2 | 0.0 | $0.0 \pm 0.0$ | 0.0 | 0.2 | 0.0 | $0.0 \pm 0.0$ | 0.0 | - |
| metaworld-box-close | 75 | 0.68 | $0.84 \pm 0.026$ | $0.99 \pm 0.017$ | 0.0 | 0.4 | 0.0 | $0.0 \pm 0.0$ | 0.0 | 0.23 | 0.0 | $0.0 \pm 0.0$ | 0.0 | - |
| metaworld-box-close | 100 | 0.81 | $0.93 \pm 0.067$ | $0.98 \pm 0.005$ | 0.0 | 0.56 | 0.0 | $0.0 \pm 0.0$ | 0.0 | 0.31 | 0.0 | $0.0 \pm 0.0$ | 0.0 | - |
| metaworld-door-lock | 25 | 0.58 | $0.73 \pm 0.019$ | $0.8 \pm 0.012$ | 0.13 | 0.33 | 0.12 | $0.12 \pm 0.012$ | 0.24 | 0.18 | 0.31 | $0.23 \pm 0.039$ | 0.0 | - |
| metaworld-door-lock | 50 | 0.55 | $0.81 \pm 0.047$ | $0.82 \pm 0.014$ | 0.13 | 0.34 | 0.15 | $0.09 \pm 0.021$ | 0.24 | 0.24 | 0.25 | $0.29 \pm 0.028$ | 0.03 | - |
| metaworld-door-lock | 75 | 0.63 | $0.85 \pm 0.028$ | $0.84 \pm 0.022$ | 0.13 | 0.39 | 0.17 | $0.14 \pm 0.029$ | 0.24 | 0.31 | 0.27 | $0.4 \pm 0.026$ | 0.01 | - |
| metaworld-door-lock | 100 | 0.64 | $0.85 \pm 0.0$ | $0.85 \pm 0.0$ | 0.13 | 0.48 | 0.17 | $0.15 \pm 0.029$ | 0.24 | 0.38 | 0.28 | $0.42 \pm 0.019$ | 0.03 | - |
| metaworld-door-unlock | 25 | 0.53 | $0.65 \pm 0.054$ | $0.72 \pm 0.037$ | 0.01 | 0.36 | 0.03 | $0.22 \pm 0.057$ | 0.06 | 0.31 | 0.07 | $0.15 \pm 0.036$ | 0.0 | - |
| metaworld-door-unlock | 50 | 0.55 | $0.81 \pm 0.049$ | $0.84 \pm 0.031$ | 0.01 | 0.47 | 0.03 | $0.23 \pm 0.08$ | 0.06 | 0.42 | 0.07 | $0.25 \pm 0.083$ | 0.01 | - |
| metaworld-door-unlock | 75 | 0.78 | $0.88 \pm 0.008$ | $0.92 \pm 0.005$ | 0.01 | 0.41 | 0.02 | $0.2 \pm 0.022$ | 0.06 | 0.44 | 0.05 | $0.3 \pm 0.074$ | 0.0 | - |
| metaworld-door-unlock | 100 | 0.86 | $0.9 \pm 0.017$ | $0.93 \pm 0.005$ | 0.01 | 0.45 | 0.03 | $0.16 \pm 0.031$ | 0.06 | 0.5 | 0.08 | $0.24 \pm 0.094$ | 0.03 | - |
| metaworld-hand-insert | 25 | 0.27 | $0.62 \pm 0.022$ | $0.54 \pm 0.022$ | 0.01 | 0.07 | 0.07 | $0.1 \pm 0.019$ | 0.01 | 0.06 | 0.05 | $0.29 \pm 0.034$ | 0.0 | - |
| metaworld-hand-insert | 50 | 0.54 | $0.68 \pm 0.031$ | $0.69 \pm 0.012$ | 0.01 | 0.13 | 0.04 | $0.16 \pm 0.031$ | 0.01 | 0.1 | 0.06 | $0.58 \pm 0.012$ | 0.0 | - |
| metaworld-hand-insert | 75 | 0.56 | $0.78 \pm 0.028$ | $0.79 \pm 0.021$ | 0.01 | 0.22 | 0.04 | $0.16 \pm 0.022$ | 0.01 | 0.1 | 0.06 | $0.63 \pm 0.017$ | 0.0 | - |
| metaworld-hand-insert | 100 | 0.69 | $0.82 \pm 0.042$ | $0.83 \pm 0.0$ | 0.01 | 0.23 | 0.02 | $0.15 \pm 0.054$ | 0.01 | 0.14 | 0.03 | $0.61 \pm 0.065$ | 0.01 | - |
| atari-pong | 7 (11599) | 0.08 | $0.08 \pm 0.0$ | $0.17 \pm 0.042$ | 0.0 | 0.0 | 0.0 | $0.0 \pm 0.0$ | 0.0 | 0.0 | 0.0 | $0.0 \pm 0.0$ | 0.0 | $0.0 \pm 0.0$ |
| atari-pong | 10 (16533) | 0.18 | $0.2 \pm 0.0$ | $0.23 \pm 0.0$ | 0.0 | 0.0 | 0.0 | $0.0 \pm 0.0$ | 0.0 | 0.0 | 0.0 | $0.0 \pm 0.005$ | 0.0 | $0.0 \pm 0.0$ |
| atari-pong | 13 (21468) | 0.18 | $0.23 \pm 0.009$ | $0.23 \pm 0.009$ | 0.0 | 0.0 | 0.0 | $0.0 \pm 0.0$ | 0.0 | 0.0 | 0.0 | $0.0 \pm 0.005$ | 0.0 | $0.04 \pm 0.016$ |
| atari-pong | 16 (26400) | 0.18 | $0.25 \pm 0.047$ | $0.3 \pm 0.005$ | 0.0 | 0.0 | 0.0 | $0.0 \pm 0.0$ | 0.0 | 0.0 | 0.0 | $0.0 \pm 0.005$ | 0.0 | $0.06 \pm 0.016$ |
| atari-alien | 3 (12759) | 0.04 | $0.08 \pm 0.005$ | $0.09 \pm 0.0$ | 0.0 | 0.0 | 0.0 | $0.0 \pm 0.0$ | 0.0 | 0.0 | 0.0 | $0.0 \pm 0.0$ | 0.0 | $0.0 \pm 0.0$ |
| atari-alien | 4 (17353) | 0.05 | $0.08 \pm 0.005$ | $0.09 \pm 0.005$ | 0.0 | 0.0 | 0.0 | $0.0 \pm 0.0$ | 0.0 | 0.0 | 0.0 | $0.0 \pm 0.0$ | 0.0 | $0.0 \pm 0.0$ |
| atari-alien | 5 (22684) | 0.05 | $0.08 \pm 0.0$ | $0.09 \pm 0.005$ | 0.0 | 0.0 | 0.0 | $0.0 \pm 0.005$ | 0.0 | 0.0 | 0.0 | $0.0 \pm 0.0$ | 0.0 | $0.0 \pm 0.0$ |
| atari-alien | 6 (27692) | 0.05 | $0.09 \pm 0.005$ | $0.1 \pm 0.005$ | 0.0 | 0.01 | 0.0 | $0.0 \pm 0.005$ | 0.0 | 0.01 | 0.0 | $0.0 \pm 0.0$ | 0.0 | $0.01 \pm 0.005$ |
| atari-mspacman | 4 (11902) | 0.68 | $0.82 \pm 0.017$ | $0.82 \pm 0.005$ | 0.0 | 0.01 | 0.0 | $0.02 \pm 0.005$ | 0.01 | 0.02 | 0.02 | $0.04 \pm 0.005$ | 0.0 | $0.1 \pm 0.026$ |
| atari-mspacman | 5 (14520) | 0.68 | $0.81 \pm 0.031$ | $0.83 \pm 0.016$ | 0.0 | 0.02 | 0.01 | $0.02 \pm 0.005$ | 0.01 | 0.02 | 0.02 | $0.04 \pm 0.0$ | 0.0 | $0.06 \pm 0.042$ |
| atari-mspacman | 9 (22490) | 0.82 | $0.85 \pm 0.029$ | $0.89 \pm 0.009$ | 0.0 | 0.02 | 0.02 | $0.03 \pm 0.005$ | 0.01 | 0.02 | 0.01 | $0.04 \pm 0.005$ | 0.0 | $0.17 \pm 0.045$ |
| atari-mspacman | 10 (25160) | 0.82 | $0.86 \pm 0.031$ | $0.9 \pm 0.0$ | 0.0 | 0.01 | 0.01 | $0.03 \pm 0.0$ | 0.01 | 0.01 | 0.02 | $0.04 \pm 0.0$ | 0.0 | $0.18 \pm 0.047$ |
| Aggregate Mean | | 0.473 | 0.602 | 0.633 | 0.019 | 0.165 | 0.029 | 0.063 | 0.04 | 0.129 | 0.052 | 0.143 | 0.004 | |

Table 1: Values in Figure 4. For a particular training seed, each agent is evaluated across 100 rollouts of different seeds in Metaworld and 15 rollouts of different seeds (with $p_{\text{sticky}} = 0.05$) in Atari. We compute the overall mean and standard deviation over three training seeds for REGENT, REGENT Finetuned, the PEFT with IA3 baselines, and DRIL. For REGENT, this means that we trained and thoroughly evaluated three REGENT policies. Then, we finetuned each of these three REGENT policies 32 times, with 8 environments and 4 points per environment, for a total of 96 finetuned policies. We compute the final average over both training seeds and evaluation seeds and the final standard deviation over training seeds. We note that R&P does not have a standard deviation across training seeds as it does not have any training parameters at all.

| Env | Num demos | R&P | REGENT | REGENT Finetuned | JAT/Gato | JAT/Gato Finetuned | JAT/Gato (All Data) | JAT/Gato (All Data) Finetuned | Train From Scratch |
|---|---|---|---|---|---|---|---|---|---|
| mujoco-halfcheetah | 25 | 0.08 | 0.01 | 0.19 | 0.0 | 0.0 | 0.0 | 0.0 | 0.04 |
| mujoco-halfcheetah | 50 | 0.08 | 0.01 | 0.26 | 0.0 | 0.01 | 0.0 | 0.0 | 0.04 |
| mujoco-halfcheetah | 75 | 0.09 | 0.0 | 0.31 | 0.0 | 0.03 | 0.0 | 0.0 | 0.04 |
| mujoco-halfcheetah | 100 | 0.1 | 0.0 | 0.34 | 0.0 | 0.03 | 0.0 | 0.01 | 0.04 |
| mujoco-hopper | 25 | 0.21 | 0.01 | 0.01 | 0.0 | 0.0 | 0.01 | 0.03 | 0.04 |
| mujoco-hopper | 50 | 0.26 | 0.01 | 0.04 | 0.0 | 0.0 | 0.01 | 0.03 | 0.02 |
| mujoco-hopper | 75 | 0.24 | 0.01 | 0.1 | 0.0 | 0.0 | 0.01 | 0.05 | 0.02 |
| mujoco-hopper | 100 | 0.25 | 0.01 | 0.12 | 0.0 | 0.0 | 0.01 | 0.05 | 0.01 |
| Aggregate Mean | | 0.164 | 0.008 | 0.171 | 0.0 | 0.009 | 0.005 | 0.021 | 0.031 |

Table 2: Values in Figure 13. Each value is an average across 100 rollouts of different seeds.

| Env | Num demos | REGENT | JAT/Gato |
|---|---|---|---|
| metaworld-assembly | 100 | 0.83 | 0.89 |
| metaworld-basketball | 100 | 0.68 | 0.03 |
| metaworld-button-press-topdown-wall | 100 | 0.62 | 0.43 |
| metaworld-button-press-topdown | 100 | 0.62 | 0.43 |
| metaworld-button-press-wall | 100 | 0.94 | 0.61 |
| metaworld-button-press | 100 | 0.62 | 0.43 |
| metaworld-coffee-button | 100 | 0.84 | 0.33 |
| metaworld-coffee-pull | 100 | 0.62 | 0.01 |
| metaworld-coffee-push | 100 | 0.18 | 0.04 |
| metaworld-dial-turn | 100 | 0.83 | 0.33 |
| metaworld-disassemble | 100 | 2.24 | 0.0 |
| metaworld-door-close | 100 | 1.0 | 0.46 |
| metaworld-door-open | 100 | 0.98 | 0.46 |
| metaworld-drawer-close | 100 | 1.0 | 0.89 |
| metaworld-drawer-open | 100 | 0.96 | 0.91 |
| metaworld-faucet-close | 100 | 0.53 | 0.16 |
| metaworld-faucet-open | 100 | 0.99 | 0.68 |
| metaworld-hammer | 100 | 0.95 | 0.52 |
| metaworld-handle-press-side | 100 | 0.99 | 0.83 |
| metaworld-handle-press | 100 | 0.99 | 0.83 |
| metaworld-handle-pull-side | 100 | 0.48 | 0.0 |
| metaworld-handle-pull | 100 | 0.48 | 0.0 |
| metaworld-lever-pull | 100 | 0.19 | 0.03 |
| metaworld-peg-insert-side | 100 | 0.7 | 0.15 |
| metaworld-peg-unplug-side | 100 | 0.31 | 0.02 |
| metaworld-pick-out-of-hole | 100 | 0.01 | 0.0 |
| metaworld-pick-place-wall | 100 | 0.99 | 0.01 |
| metaworld-pick-place | 100 | 0.99 | 0.01 |
| metaworld-plate-slide-back-side | 100 | 1.0 | 0.02 |
| metaworld-plate-slide-back | 100 | 1.0 | 0.02 |
| metaworld-plate-slide-side | 100 | 0.99 | 0.19 |
| metaworld-plate-slide | 100 | 1.0 | 0.02 |
| metaworld-push-back | 100 | 0.84 | 0.01 |
| metaworld-push-wall | 100 | 0.81 | 0.01 |
| metaworld-push | 100 | 0.84 | 0.01 |
| metaworld-reach-wall | 100 | 0.99 | 1.03 |
| metaworld-reach | 100 | 0.99 | 1.03 |
| metaworld-shelf-place | 100 | 0.96 | 0.0 |
| metaworld-soccer | 100 | 0.61 | 0.21 |
| metaworld-stick-pull | 100 | 0.88 | 0.04 |
| metaworld-stick-push | 100 | 0.75 | 0.02 |
| metaworld-sweep-into | 100 | 0.91 | 0.16 |
| metaworld-sweep | 100 | 0.91 | 0.16 |
| metaworld-window-close | 100 | 0.9 | 0.1 |
| metaworld-window-open | 100 | 0.97 | 0.14 |
| Aggregate Mean | | 0.82 | 0.281 |

Table 3: Values in Figure 15. Each value is an average across 100 rollouts of different seeds.

| Env | Num demos | REGENT | JAT/Gato |
|---|---|---|---|
| mujoco-ant | 100 | 0.17 | 0.12 |
| mujoco-doublependulum | 100 | 0.02 | 0.01 |
| mujoco-humanoid | 100 | 0.02 | 0.02 |
| mujoco-pendulum | 100 | 0.06 | 0.03 |
| mujoco-pusher | 100 | 0.9 | 0.95 |
| mujoco-reacher | 100 | 0.9 | 0.91 |
| mujoco-standup | 100 | 0.26 | 0.2 |
| mujoco-swimmer | 100 | 0.82 | 0.97 |
| mujoco-walker | 100 | 0.05 | 0.16 |
| Aggregate Mean | | 0.356 | 0.374 |

Table 4: Values in Figure 16. Each value is an average across 100 rollouts of different seeds.

| Env | Num demos | REGENT | JAT/Gato |
|---|---|---|---|
| atari-amidar | 5 | 0.96 | 0.02 |
| atari-assault | 5 | 0.02 | 0.02 |
| atari-asterix | 5 | 0.13 | 0.03 |
| atari-asteroids | 5 | 0.07 | 0.0 |
| atari-atlantis | 5 | 0.18 | 0.03 |
| atari-bankheist | 5 | 0.46 | 0.18 |
| atari-battlezone | 5 | 0.47 | 0.01 |
| atari-beamrider | 5 | 0.04 | 0.0 |
| atari-berzerk | 5 | 0.14 | 0.0 |
| atari-bowling | 5 | 1.0 | 1.0 |
| atari-boxing | 5 | 0.22 | 0.38 |
| atari-breakout | 5 | 0.15 | 0.0 |
| atari-centipede | 5 | 0.35 | 0.07 |
| atari-choppercommand | 5 | 0.02 | 0.0 |
| atari-crazyclimber | 5 | 0.19 | 0.14 |
| atari-defender | 5 | 0.1 | 0.01 |
| atari-demonattack | 5 | 0.0 | 0.0 |
| atari-doubledunk | 5 | 0.54 | 0.28 |
| atari-enduro | 5 | 0.0 | 0.0 |
| atari-fishingderby | 5 | 0.62 | 0.1 |
| atari-freeway | 5 | 0.82 | 0.6 |
| atari-frostbite | 5 | 0.44 | 0.01 |
| atari-gopher | 5 | 0.66 | 0.01 |
| atari-gravitar | 5 | 0.03 | 0.06 |
| atari-hero | 5 | 1.0 | 0.17 |
| atari-icehockey | 5 | 0.21 | 0.1 |
| atari-jamesbond | 5 | 0.01 | 0.0 |
| atari-kangaroo | 5 | 0.0 | 0.0 |
| atari-krull | 5 | 0.5 | 0.41 |
| atari-kungfumaster | 5 | 0.0 | 0.02 |
| atari-montezumarevenge | 5 | 0.0 | 0.0 |
| atari-namethisgame | 5 | 0.61 | 0.03 |
| atari-phoenix | 5 | 0.07 | 0.0 |
| atari-pitfall | 5 | 0.97 | 1.01 |
| atari-privateeye | 5 | 0.0 | 0.79 |
| atari-qbert | 5 | 0.57 | 0.01 |
| atari-riverraid | 5 | 0.19 | 0.04 |
| atari-roadrunner | 5 | 0.03 | 0.02 |
| atari-robotank | 5 | 0.23 | 0.04 |
| atari-seaquest | 5 | 0.48 | 0.11 |
| atari-skiing | 5 | 0.0 | 0.0 |
| atari-solaris | 5 | 0.0 | 0.0 |
| atari-surround | 5 | 0.36 | 0.03 |
| atari-tennis | 5 | 0.15 | 0.01 |
| atari-timepilot | 5 | 0.04 | 0.0 |
| atari-tutankham | 5 | 0.16 | 0.02 |
| atari-upndown | 5 | 0.13 | 0.01 |
| atari-venture | 5 | 1.0 | 1.0 |
| atari-videopinball | 5 | 0.0 | 0.01 |
| atari-wizardofwor | 5 | 0.76 | 0.01 |
| atari-yarsrevenge | 5 | 0.01 | 0.01 |
| atari-zaxxon | 5 | 0.14 | 0.01 |
| Aggregate Mean | | 0.293 | 0.131 |

Table 5: Values in Figure 17. Each value is an average across 15 rollouts of different seeds.

| Env | Num demos | REGENT | JAT/Gato |
|---|---|---|---|
| babyai-action-obj-door | 20 | 0.39 | 0.57 |
| babyai-blocked-unlock-pickup | 20 | 0.45 | 0.97 |
| babyai-boss-level-no-unlock | 20 | 0.49 | 0.1 |
| babyai-boss-level | 20 | 0.49 | 0.1 |
| babyai-find-obj-s5 | 20 | 0.61 | 0.92 |
| babyai-go-to-door | 20 | 0.49 | 0.89 |
| babyai-go-to-imp-unlock | 20 | 0.21 | 0.03 |
| babyai-go-to-local | 20 | 0.66 | 0.46 |
| babyai-go-to-obj-door | 20 | 0.14 | 0.76 |
| babyai-go-to-obj | 20 | 0.14 | 0.76 |
| babyai-go-to-red-ball-grey | 20 | 0.68 | 0.76 |
| babyai-go-to-red-ball-no-dists | 20 | 0.95 | 0.98 |
| babyai-go-to-red-ball | 20 | 0.68 | 0.76 |
| babyai-go-to-red-blue-ball | 20 | 0.71 | 0.71 |
| babyai-go-to-seq | 20 | 0.46 | 0.46 |
| babyai-go-to | 20 | 0.49 | 0.89 |
| babyai-key-corridor | 20 | 0.07 | 0.47 |
| babyai-mini-boss-level | 20 | 0.61 | 0.28 |
| babyai-move-two-across-s8n9 | 20 | 0.0 | 0.0 |
| babyai-one-room-s8 | 20 | 0.96 | 0.97 |
| babyai-open-door | 20 | 0.58 | 0.92 |
| babyai-open-doors-order-n4 | 20 | 0.54 | 0.33 |
| babyai-open-red-door | 20 | 0.99 | 0.99 |
| babyai-open-two-doors | 20 | 0.37 | 0.37 |
| babyai-open | 20 | 0.58 | 0.92 |
| babyai-pickup-above | 20 | 0.44 | 0.86 |
| babyai-pickup-dist | 20 | 0.66 | 0.4 |
| babyai-pickup-loc | 20 | 0.42 | 0.38 |
| babyai-pickup | 20 | 0.44 | 0.86 |
| babyai-put-next-local | 20 | 0.12 | 0.0 |
| babyai-put-next | 20 | 0.12 | 0.0 |
| babyai-synth-loc | 20 | 0.5 | 0.3 |
| babyai-synth-seq | 20 | 0.59 | 0.07 |
| babyai-synth | 20 | 0.5 | 0.3 |
| babyai-unblock-pickup | 20 | 0.25 | 0.23 |
| babyai-unlock-local | 20 | 0.93 | 0.99 |
| babyai-unlock-pickup | 20 | 0.81 | 0.96 |
| babyai-unlock-to-unlock | 20 | 0.35 | 0.79 |
| babyai-unlock | 20 | 0.93 | 0.99 |
| Aggregate Mean | | 0.508 | 0.577 |

Table 6: Values in Figure 18. Each value is an average across 100 rollouts of different seeds.

# D  ADDITIONAL PROCGEN RESULTS

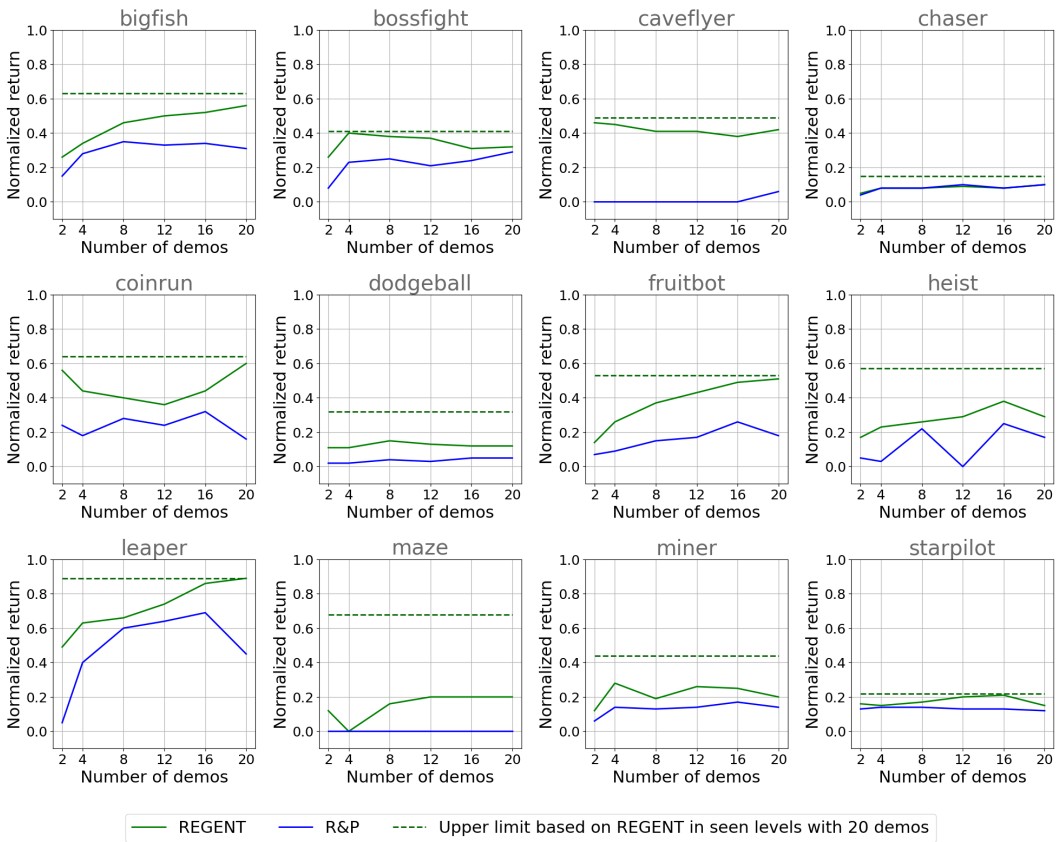

Figure 19: Normalized returns in unseen levels in all 12 ProcGen training environments against the number of demonstration trajectories the agent can retrieve from. Each value is an average across 10 levels with 5 rollouts each with $p_{\text{sticky}} = 0.1$. See Table 8 for all values.

| Env | sticky prob | num demos | (epoch, end batch) | REGENT | R&P | MTT [37] |
|---|---|---|---|---|---|---|
| ninja | p=0.2 | n=2 | (0, 11612) | 7.13± 0.09 | 5.67 | |
| | | n=4 | (0, 11612) | 7.53± 0.34 | 5.5 | 4.1± 0.35 |
| | | n=8 | (0, 11612) | 6.8± 0.85 | 6.4 | |
| | | n=12 | (0, 11612) | 7.6± 0.28 | 6.33 | |
| | | n=16 | (0, 11612) | 7.8± 0.28 | 7.4 | |
| | | n=20 | (0, 11612) | 8.07± 0.19 | 7.6 | |
| climber | p=0.2 | n=2 | (0, 11612) | 4.65± 0.25 | 3.17 | |
| | | n=4 | (0, 11612) | 6.01± 0.52 | 3.95 | 1.85± 0.41 |
| | | n=8 | (0, 11612) | 5.89± 0.44 | 3.12 | |
| | | n=12 | (0, 11612) | 7.02± 0.13 | 5.64 | |
| | | n=16 | (0, 11612) | 6.2± 0.43 | 3.77 | |
| | | n=20 | (0, 11612) | 6.13± 0.72 | 5.41 | |
| plunder | p=0.2 | n=2 | (0, 11612) | 10.48± 0.43 | 8.46 | |
| | | n=4 | (0, 11612) | 12.33± 0.48 | 8.3 | 2.5± 0.2 |
| | | n=8 | (0, 11612) | 12.23± 0.66 | 8.2 | |
| | | n=12 | (0, 11612) | 13.51± 0.22 | 9.36 | |
| | | n=16 | (0, 11612) | 13.75± 0.2 | 9.84 | |
| | | n=20 | (0, 11612) | 14.13± 0.26 | 10.8 | |
| jumper | p=0.2 | n=2 | (0, 11612) | 4.53± 0.25 | 3.75 | |
| | | n=4 | (0, 11612) | 5.4± 0.57 | 4.0 | 4.65± 0.28 |
| | | n=8 | (0, 11612) | 5.6± 0.71 | 4.25 | |
| | | n=12 | (0, 11612) | 6.47± 1.06 | 5.25 | |
| | | n=16 | (0, 11612) | 6.47± 0.47 | 6.0 | |
| | | n=20 | (0, 11612) | 6.4± 0.33 | 5.5 | |

Table 7: Values in Figure 5 before normalization. We trained three REGENT generalist agents for three training seeds. We evaluated each of the three agents' performance, as well as the R&P agent, across 10 levels with 5 rollouts and $p_{\text{sticky}} = 0.2$ in each unseen ProcGen environment. We computed the final average over both training seeds and evaluation levels/rollouts and the final standard deviation over training seeds. The values for MTT are the best scores reported in [37].

| Env | sticky prob | num demos | (epoch, end batch) | REGENT | R&P | REGENT (ID seed) |
|---|---|---|---|---|---|---|
| bigfish | p=0.1 | n=2 | (0, 11612) | 11.32 | 6.75 | |
| | | n=4 | (0, 11612) | 14.32 | 11.85 | |
| | | n=8 | (0, 11612) | 18.82 | 14.7 | |
| | | n=12 | (0, 11612) | 20.66 | 14.02 | |
| | | n=16 | (0, 11612) | 21.32 | 14.43 | |
| | | n=20 | (0, 11612) | 22.74 | 13.28 | 25.7 |
| bossfight | p=0.1 | n=2 | (0, 11612) | 3.76 | 1.45 | |
| | | n=4 | (0, 11612) | 5.52 | 3.38 | |
| | | n=8 | (0, 11612) | 5.24 | 3.67 | |
| | | n=12 | (0, 11612) | 5.12 | 3.17 | |
| | | n=16 | (0, 11612) | 4.32 | 3.51 | |
| | | n=20 | (0, 11612) | 4.44 | 4.1 | 5.6 |
| caveflyer | p=0.1 | n=2 | (0, 11612) | 7.44 | 2.74 | |
| | | n=4 | (0, 11612) | 7.34 | 2.98 | |
| | | n=8 | (0, 11612) | 7.0 | 3.29 | |
| | | n=12 | (0, 11612) | 6.96 | 3.27 | |
| | | n=16 | (0, 11612) | 6.72 | 3.31 | |
| | | n=20 | (0, 11612) | 7.08 | 4.04 | 7.68 |
| chaser | p=0.1 | n=2 | (0, 11612) | 1.15 | 1.01 | |
| | | n=4 | (0, 11612) | 1.51 | 1.51 | |
| | | n=8 | (0, 11612) | 1.46 | 1.47 | |
| | | n=12 | (0, 11612) | 1.61 | 1.77 | |
| | | n=16 | (0, 11612) | 1.47 | 1.53 | |
| | | n=20 | (0, 11612) | 1.79 | 1.73 | 2.4 |
| coinrun | p=0.1 | n=2 | (0, 11612) | 7.8 | 6.2 | |
| | | n=4 | (0, 11612) | 7.2 | 5.9 | |
| | | n=8 | (0, 11612) | 7.0 | 6.4 | |
| | | n=12 | (0, 11612) | 6.8 | 6.2 | |
| | | n=16 | (0, 11612) | 7.2 | 6.6 | |
| | | n=20 | (0, 11612) | 8.0 | 5.8 | 8.2 |
| dodgeball | p=0.1 | n=2 | (0, 11612) | 3.36 | 1.93 | |
| | | n=4 | (0, 11612) | 3.44 | 1.8 | |
| | | n=8 | (0, 11612) | 4.04 | 2.17 | |
| | | n=12 | (0, 11612) | 3.72 | 2.07 | |
| | | n=16 | (0, 11612) | 3.64 | 2.33 | |
| | | n=20 | (0, 11612) | 3.56 | 2.4 | 7.12 |
| fruitbot | p=0.1 | n=2 | (0, 11612) | 3.18 | 0.98 | |
| | | n=4 | (0, 11612) | 7.48 | 1.6 | |
| | | n=8 | (0, 11612) | 11.02 | 3.67 | |
| | | n=12 | (0, 11612) | 13.02 | 4.38 | |
| | | n=16 | (0, 11612) | 15.04 | 7.45 | |
| | | n=20 | (0, 11612) | 15.62 | 4.67 | 16.36 |
| heist | p=0.1 | n=2 | (0, 11612) | 4.6 | 3.8 | |
| | | n=4 | (0, 11612) | 5.0 | 3.7 | |
| | | n=8 | (0, 11612) | 5.2 | 4.9 | |
| | | n=12 | (0, 11612) | 5.4 | 3.4 | |
| | | n=16 | (0, 11612) | 6.0 | 5.1 | |
| | | n=20 | (0, 11612) | 5.4 | 4.6 | 7.2 |
| leaper | p=0.1 | n=2 | (0, 11612) | 6.4 | 3.33 | |
| | | n=4 | (0, 11612) | 7.4 | 5.83 | |
| | | n=8 | (0, 11612) | 7.6 | 7.17 | |
| | | n=12 | (0, 11612) | 8.2 | 7.5 | |
| | | n=16 | (0, 11612) | 9.0 | 7.83 | |
| | | n=20 | (0, 11612) | 9.2 | 6.17 | 9.2 |
| maze | p=0.1 | n=2 | (0, 11612) | 5.6 | 5.0 | |
| | | n=4 | (0, 11612) | 4.8 | 3.1 | |
| | | n=8 | (0, 11612) | 5.8 | 4.9 | |
| | | n=12 | (0, 11612) | 6.0 | 4.6 | |
| | | n=16 | (0, 11612) | 6.0 | 4.5 | |
| | | n=20 | (0, 11612) | 6.0 | 4.5 | 8.4 |
| miner | p=0.1 | n=2 | (0, 11612) | 2.88 | 2.14 | |
| | | n=4 | (0, 11612) | 4.72 | 3.12 | |
| | | n=8 | (0, 11612) | 3.68 | 2.96 | |
| | | n=12 | (0, 11612) | 4.54 | 3.09 | |
| | | n=16 | (0, 11612) | 4.38 | 3.45 | |
| | | n=20 | (0, 11612) | 3.76 | 3.12 | 6.58 |
| starpilot | p=0.1 | n=2 | (0, 11612) | 12.52 | 10.28 | |
| | | n=4 | (0, 11612) | 11.94 | 11.33 | |
| | | n=8 | (0, 11612) | 12.82 | 10.93 | |
| | | n=12 | (0, 11612) | 14.9 | 10.8 | |
| | | n=16 | (0, 11612) | 15.34 | 10.49 | |
| | | n=20 | (0, 11612) | 11.8 | 10.08 | 16.1 |

Table 8: Values in Figures 6 and 19 before normalization. Each value is an average across 10 levels with 5 rollouts each.

# E  ADDITIONAL RELATED WORK

In addition to our related work section, we discuss other relevant papers where a retrieval bias has been recognized as a useful component here [61, 57, 59, 23, 25, 39].

RAEA [61] performs behavior cloning in each new environment with access to a "policy memory bank" and focuses on improving performance in a training task. In RAEA's appendix, the authors demonstrate very initial signs of generalization to new tasks only after training on a few demonstrations from the new tasks. REGENT on the other hand pretrains a generalist policy that can generalize without any finetuning to completely new environments with different observations, dynamics, and rewards. Re-ViLM [57] trains a retrieval-augmented image captioning model, which while demonstrating the usefulness of a retrieval bias beyond robotics and game-playing tasks, is not applicable to our settings.

Expel [59] and RAP [23] build LLM agents for high-level planning on top of frozen LLMs. Although off-the-shelf LLMs are not yet helpful with low-level fine-grained continuous control, these methods could be key in high-level semantic text-based action spaces.

GLAs [25] and RADT [39] are recent in-context reinforcement learning (RL) methods. We briefly discussed earlier in-context RL methods in our related work section. GLAs attempt to generalize to new mujoco embodiments via in-context RL but only show minor improvements over random agents. RADT emphasizes that they are unable to generalize via in-context RL to unseen procgen, metaworld, or DMControl environments. REGENT, on the other hand, is an in-context imitation learning method that generalizes to unseen metaworld, atari, and procgen environments via retrieval augmentation (from a retrieval buffer of a few expert demonstrations) and in-context learning. REGENT also struggles with generalization to new mujoco embodiments like GLAs.

