# OpenReview forum: "REGENT: A Retrieval-Augmented Generalist Agent That Can Act In-Context in New Environments"
_ICLR.cc/2025/Conference — ICLR 2025 Oral_

### Official Review · Reviewer_8qsk · 2024-10-30

**Soundness:** 3
**Presentation:** 2
**Contribution:** 3
**Rating:** 8
**Confidence:** 5

**Summary:**

This work proposes to leverage retrieval augmentation to reduce the amount of training data required for training generalist agents.
The authors introduce $\texttt{REGENT}$, a retrieval-augmented architecture that incorporates a non-parametric retrieval component (R&P) with learning of a parametric policy.
Further, they demonstrate theoretical guarantees for performance with respect to the number of expert demonstrations.
They pre-train $\texttt{REGENT}$ on 4 different domains and evaluate on unseen levels and holdout tasks, where it exhibits strong performance compared to baselines.

**Strengths:**

The authors provide theoretical guarantees for performance with respect to the number of demonstrations.

$\texttt{REGENT}$ is a novel architecture that effectively incorporates retrieval augmentation for imitation learning.

Retrieval augmentation enables training on much less data than other generalist agents.

The authors demonstrate the promise of retrieval augmentation for unseen levels and tasks.

**Weaknesses:**

**Significance of results:**

The authors only provide averages in all figures (except for Figure 7), but no variance estimates. How do those look? Are the results on the unseen tasks statistically significant?

The authors claim that $\texttt{REGENT}$ outperforms JAT even after fine-tuning, but there is little information about the fine-tuning protocol, i.e. what fine-tuning technique is used, etc. Therefore it is difficult to determine the significance of the results. Can the authors elaborate a on this?
Also there are fine-tuning techniques specifically designed for few-shot learning, which may be more suitable in this setting [1,2].

On unseen levels of ProcGen the authors only compare $\texttt{REGENT}$ to R&P, but no results for MTT, why is this the case?
BC is a very weak baseline for training from scratch. The authors should compare to other more recent imitation learning techniques [3,4,5] that use the same amount of expert demonstrations as $\texttt{REGENT}$. This would shed light on the significance of the reported results.

Another interesting experiment would be zero-shot evaluation without having expert demonstrations on the unseen tasks. If $\texttt{REGENT}$ performs well in this scenario, this would greatly strengthen the paper.

There is a lack of ablation studies, for example performance on varying the number of examples in the context and ablation on different distance metrics.

Finally, it is not really clear why the authors differentiate between the two different settings. Why not include ProcGen in the first setting?
Why are different baselines used for the different settings? MTT could also be applied to the first setting, right?
At least having a fixed set of all baselines across both settings would strengthen the paper.

**Limitations and claims**

- The authors claim that deployment on unseen tasks is instantaneously, this is not true though, as deployment in an unseen environment requires collection of expert demonstrations which in turn requires training of an expert policy. This should be made more explicit as a limitation of the proposed method.
- Moreover, claims on "handful of examples" should be caveated as thousands of samples are not a handful.
- Authors claim apples-to-apples comparison between JAT and $\texttt{REGENT}$ when being trained on the same amount of data. This is not apples-to-apples though as $\texttt{REGENT}$ uses expert demonstrations of the unseen tasks. The more fair comparison is to fine-tuning JAT.
- Further, the work would benefit from a bit more accurate positioning on the landscape of methods that do or do not rely on demonstrations. This is an important aspect, as there are approaches that aim at leveraging in-context learning in the context of learning-to-learn as in meta-learning [6]. Examples are [7,8], where the latter also relies on retrieval augmentation.

**Presentation:**

Presentation can be improved, in particular:

* Eq. 1 & 2: $c_t$ is provided as input to the policy, but it is never used, it is only used in Eq. 3, as R&P does not rely on context information
* Figure 3: enumeration in caption, but not in figure, either remove them in caption or add in figure
* Paragraph on $\texttt{REGENT}$ architecture is not really clear as the reader does not have an overview what modalities are represented in the training data, therefore the choice of encodings seem arbitrary.
* Theorem 5.2: $\boldsymbol{H}$ represents the horizon, right? In section 3 it is not boldface, this should be consistent.
* Figure 8: hard to follow what actions are "good" or "bad" as there is no explanation for the action index

**References:**

[1] Few-Shot Parameter-Efficient Fine-Tuning is Better and Cheaper than In-Context Learning, Liu et al., NeurIPS 2022

[2] Scaling & Shifting Your Features: A New Baseline for Efficient Model Tuning, Lian et al., NeurIPS 2022

[3] Generative Adversarial Imitation Learning, Ho et al., NeurIPS 2016

[4] Disagreement-Regularized Imitation Learning, Brantley et  al., ICLR 2020

[5] Of Moments and Matching: A Game-Theoretic Framework for Closing the Imitation Gap, Swamy et al., ICML 2021

[6]   Evolutionary principles in self-referential learning, or on learning how to learn, Schmidhuber et al., Diploma Thesis, 1987

[7] Towards General-Purpose In-Context Learning Agents, Kirsch et al., NeurIPS 2023 FMDM Workshop

[8] Retrieval-Augmented Decision Transformer: External Memory for In-context RL, Schmied et al., arXiv 2024

**Questions:**

- Why is the scaling factor for MixedReLU set to 10? This seems like an arbitrary choice, why not remove it entirely?
- Does the number of demonstrations for MTT on ProcGen match the number of demonstrations for $\texttt{REGENT}$?
- Is it always the case that $\texttt{REGENT}$ only receives the current query state in its context? Have you tried providing more sequential information?
- What ProcGen mode is being used? Easy or hard?
- How important is the ordering in the context of $\texttt{REGENT}$? Did you try permuting the order?
- In Figure 15 in Appendix D not all curves show improvement for using more demonstrations, do the authors have an intuition why this is the case?
- From Figures 11, 12, 13, 14 it can be seen that JAT is particularly good on BabyAI, but not good on other domains, while $\texttt{REGENT}$ is very good at Metaworld. Do the authors have an intuition why this is the case?

---

> ### Author Response · Authors · 2024-11-20
> **Official Comment by Authors (Part 1 of 4)**
>
> We thank the reviewer for the detailed review, insightful feedback, and helpful suggestions. Please let us know if you have lingering questions and whether we can provide any additional clarifications during the discussion period to improve your rating of our paper.
>
> > The authors only provide averages in all figures (except for Figure 7), but no variance estimates. How do those look? Are the results on the unseen tasks statistically significant?
>
> We note that prior work including Gato does not provide standard deviations because doing so requires training multiple generalist policies and is very compute expensive and there is good reason to expect the standard deviations to be small. For the same reasons, we did not initially do so.
>
> **We have now updated Figure 4 in the paper with the standard deviations for ‘REGENT’ and ‘REGENT Finetuned’.** They continue to demonstrate that REGENT and REGENT Finetuned are significantly better than all the baselines.
>
> To obtain this standard deviation for ‘REGENT’ in Figure 4, we trained three REGENT generalist agents for three different seeds. We evaluated each of the three agents’ performance (like before) on each unseen environment for each point on 100 rollouts of different seeds in metaworld and 15 rollouts of different seeds in atari. We compute the final average over both training seeds and evaluation seeds and the final standard deviation over training seeds.
>
> To obtain the standard deviation for ‘REGENT Finetuned’, in Figure 4, we trained 96 finetuned models (i.e., 3 pretrained agents x 8 envs x 4 points in each env). Each finetuned model is evaluated as before on 100 rollouts of different seeds in metaworld and 15 rollouts of different seeds in atari. We take the mean across model training seeds and evaluation seeds, the standard deviation across model training seeds. We have also added the exact standard deviation values in Table 1.
>
>
>
> All JAT/Gato baselines have performance at random (even after finetuning) in most environments and require more than a week to train the model. R&P, on the other hand, does not have a direct method to calculate standard deviation across training seeds as it does not have any training parameters at all. We are now training ProcGen generalist agents (which can take up to 4 days to train) for different seeds and we will try to update their standard deviations as soon as they are done either here or in the camera ready.
>
> > The authors claim that REGENT outperforms JAT even after fine-tuning, but there is little information about the fine-tuning protocol, i.e. what fine-tuning technique is used, etc. Therefore it is difficult to determine the significance of the results. Can the authors elaborate a on this? Also there are fine-tuning techniques specifically designed for few-shot learning, which may be more suitable in this setting [1,2].
>
> We refer the reviewer to Appendix A under "Finetuning hyperparameters" where we described the finetuning details and hyperparameters. We finetune the whole model for a few epochs starting from a reduced learning rate (compred to the pretraining learning rate) like in the multi-game decision transformer paper [Lee at al]. We use the same finetuning recipe for both REGENT and JAT/Gato.
>
> Also, thank you for the suggestion. These methods [1,2] unfortunately focus only on parameter-efficient finetuning for large models, where full finetuning is not possible with limited compute. Both REGENT and JAT/Gato are relatively small models that can be fully finetuned. Full finetuning has been shown to be much better than any parameter-efficient finetuning method [Shuttleworth et al].
>
> [Shuttleworth et al] Shuttleworth et al. LoRA vs Full Fine-tuning: An Illusion of Equivalence, arxiv
>
> [Lee et al] Lee et al. Multi Game Decision Transformers, NeurIPS 22

---

> ### Author Response · Authors · 2024-11-20
> **Official Comment by Authors (Part 2 of 4)**
>
> > On unseen levels of ProcGen the authors only compare REGENT to R&P, but no results for MTT, why is this the case? BC is a very weak baseline for training from scratch.
>
> > Finally, it is not really clear why the authors differentiate between the two different settings. Why not include ProcGen in the first setting? Why are different baselines used for the different settings? MTT could also be applied to the first setting, right? At least having a fixed set of all baselines across both settings would strengthen the paper.
>
> The MTT model weights are closed-source. In their ICML publication, the authors do not provide results on new levels. MTT is also difficult to train from scratch in any setting since they put multiple full trajectories in the context of the transformer, leading to exploding GPU VRAM requirements beyond academic compute resources.
>
> However, we simply report the best values from their paper in their exact setting to demonstrate REGENT's significantly improved performance even with a 3x smaller model and an order-of-magnitude less pretraining data.
>
> Similarly, we compare with JAT/Gato in their exact setting and significantly outperform JAT/Gato and JAT/Gato Finetuned, even though JAT/Gato has 1.5x our parameters and 5-10x our data.
>
> We updated our paper in Appendix A under “Additional information on MTT” to state all of the above clearly.
>
> > Does the number of demonstrations for MTT on ProcGen match the number of demonstrations for REGENT?
>
> As mentioned in the legend of Figure 4, we plotted the MTT result when it has 4 full demonstration trajectories in its context. These are the best results reported in their paper. REGENT, even with just 2 demonstrations to retrieve from, outperforms MTT's best result.
>
>
> > The authors should compare to other more recent imitation learning techniques [3,4,5] that use the same amount of expert demonstrations as REGENT. This would shed light on the significance of the reported results.
>
> We thank the reviewer for the suggestions. GAIL [3] has been shown to struggle with image observations [4], a key observation modality for our generalist agent. In [4], GAIL fails to learn an environment specific policy for atari-pong and atari-mspacman, only obtaining random agent performance in these two environments. The moment matching variants in [5] are also focused on vector observation environments. Also in [4], DRIL reports results without a sticky probability which is crucial to test generalization under stochastic dynamics. We are now retraining DRIL so that we can evaluate it, like REGENT, with a sticky probability in atari environments and we will share these results or similar results with a more recent imitation learning method as soon as possible.
>
> Finally, we would like to highlight that REGENT, unlike any imitation learning method, does not need to train on data in new environments and can generalize simply via in-context learning.
>
> > Another interesting experiment would be zero-shot evaluation without having expert demonstrations on the unseen tasks. If REGENT performs well in this scenario, this would greatly strengthen the paper.
>
> We thank the reviewer for the suggestion. But, we require at least one retrieved state, reward, action for REGENT's interpolation between R&P and the output of the transformer. Our formulation is meant specifically for few-shot (at least 1 shot) in-context learning.
>
> > There is a lack of ablation studies, for example performance on varying the number of examples in the context and ablation on different distance metrics.
>
> Thank you for the suggestion. **We have now added these new results to Appendix C.** We plotted the normalized return obtained by REGENT for different context sizes at evaluation time in unseen environments in Figure 9. We find that, in general, a larger context size leads to better performance. We also notice that the improvement in performance saturates quickly in many environments. Our choice of 20 states in the context (including the query state) leads to the best performance overall.
>
> On distance metrics: the resnet18 embedding model in the JAT/Gato setting and the L2 distance were simple choices that we tried. Using dalle based embeddings (instead of the resnet18 model) did not work and produced a REGENT policy that could not generalize better than random to new atari environments. We speculate that the imagenet trained resnet18 provided more general features transferable to atari games than the dalle encoder trained on artistic images.

---

> ### Author Response · Authors · 2024-11-20
> **Official Comment by Authors (Part 3 of 4)**
>
> > The authors claim that deployment on unseen tasks is instantaneously, this is not true though, as deployment in an unseen environment requires collection of expert demonstrations which in turn requires training of an expert policy. This should be made more explicit as a limitation of the proposed method.
>
> We thank the author for this helpful suggestion. We have rephrased the corresponding sentence in the introduction of the paper from “REGENT permits near-instantaneous deployment in unseen environments…” to “REGENT permits direct deployment in unseen environments…”. We have made the same change to a sentence in the conclusion section as well.
>
> > Moreover, claims on "handful of examples" should be caveated as thousands of samples are not a handful.
>
> This is just a misunderstanding. In Line 166 in the updated and initial pdf, we said “handful of them” where “them” refers to “demonstrations”. This is not intended to suggest we are only using a handful of transitions. We mean a handful of demonstration trajectories. For example, in the unseen atari environments in Figure 4, almost all results were obtained with less than 10 demonstrations. We refer the reviewer to Table 1 in Appendix C for a conversion from number of states to number of demonstrations. In the unseen procgen environments and levels in Figure 5 and 6, we use {2, 4, 8, 12, 16, 20} demonstrations (please see the x axis of the plots).
>
> > Authors claim apples-to-apples comparison between JAT and REGENT when being trained on the same amount of data. This is not apples-to-apples though as REGENT uses expert demonstrations of the unseen tasks. The more fair comparison is to fine-tuning JAT.
>
> This is just a misunderstanding. In Line 373 in the updated pdf and Line 406 in the initial pdf in Section 6, we used the term "apples-to-apples" to only denote that both REGENT and JAT/Gato were trained on the same data while the 'JAT/Gato (All Data)' baseline was trained on a 10x larger dataset.
>
> As the reviewer mentioned, we already plotted the results of finetuned JAT/Gato and finetuned JAT/Gato (All Data) in Figure 5. We also already described in Section 6 (under the "Effect of Finetuning"), that REGENT, with only retrieval and no finetuning, can significantly outperform both finetuned JAT/Gato and finetuned JAT/Gato (All Data).
>
> > Further, the work would benefit from a bit more accurate positioning on the landscape of methods that do or do not rely on demonstrations. This is an important aspect, as there are approaches that aim at leveraging in-context learning in the context of learning-to-learn as in meta-learning [6, Schmidhuber’s diploma thesis]. Examples are [7, Towards General Learning Agents (GLA)],[8, Retrieval Augmented Decision Transformer (RADT)], where the latter also relies on retrieval augmentation.
>
> GLAs [7] and RADT [8] are recent in-context reinforcement learning (RL) methods. We briefly discussed earlier in-context RL methods in our related work section. GLAs [7] attempt to generalize to new mujoco embodiments via in-context RL but only show minor improvements over random agents. RADT [8] emphasizes that they are unable to generalize via in-context RL to unseen procgen, metaworld, or DMControl environments.
>
> REGENT, on the other hand, is an in-context imitation learning method that generalizes to unseen metaworld, atari, and procgen environments via retrieval augmentation (from a retrieval buffer of a few expert demonstrations) and in-context learning. REGENT also struggles with generalization to new mujoco embodiments like GLAs (see Lines 467-470 in the updated pdf and Lines 471-474 in the initial pdf). **We have added the above discussion to the extended related work in the newly added Appendix E in our updated pdf.**
>
> Please also note that RADT [8] was posted on arxiv on Oct 09 2024, which was 8 days after the ICLR conference deadline.
>
> > Eq. 1 & 2: ct is provided as input to the policy, but it is never used, it is only used in Eq. 3, as R&P does not rely on context information
>
> We would like to clarify that the first retrieved action in the context (c_t) is what R&P outputs and the hence context is essential.
>
> Yet, we have cleaned up the complete subsections 4.1 and 4.2 in the new version of the paper and we thank the reviewer for the suggestion.
>
> > Paragraph on REGENT architecture is not really clear as the reader does not have an overview what modalities are represented in the training data, therefore the choice of encodings seem arbitrary.
>
> We discuss the modalities in each environment suite in Section 4 under “REGENT Training Data and Environment Suites”. We are happy to move this above the architecture if the reviewer believes that will help readers.
>
> > Theorem 5.2: H represents the horizon, right? In section 3 it is not boldface, this should be consistent.
>
> Yes. Thank you. We removed the boldface from theorem 5.2 which was unnecessary.

---

> > ### Author Response · Authors · 2024-11-20
> > **Official Comment by Authors (Part 4 of 4)**
> >
> > > Figure 8: hard to follow what actions are "good" or "bad" as there is no explanation for the action index
> >
> > Thank you for the question. We simply presented two example action predictions from REGENT in Figure 8. In one prediction (in the blue box on the right), REGENT predicts the same action as R&P (first retrieved action). In another prediction (in the black box on the left), REGENT predicts a completely different action. We simply speculate that these differences in actions with R&P, a consequence of REGENT’s in-context learning abilities, lead to higher returns at the end of a rollout for REGENT as compared with R&P.
> >
> > > Why is the scaling factor for MixedReLU set to 10? This seems like an arbitrary choice, why not remove it entirely?
> >
> > The outputs of MixedReLU are between -1 and 1, but the actions in the continuous robotics environments are not normalized. Hence, we simply set L to be a value greater than any action dimension in the pretraining data. This value turned out to be 10.
> >
> > > Is it always the case that REGENT only receives the current query state in its context? Have you tried providing more sequential information?
> >
> > The state can be a sequence of images or vectors. In atari for example, as shown in Figure 8 and described in Section 4 under "REGENT Training Data and Environment Suites", each state is already window of four grayscale images.
> >
> > > What ProcGen mode is being used? Easy or hard?
> >
> > We use Easy following MTT.
> >
> > > How important is the ordering in the context of REGENT? Did you try permuting the order?
> >
> > Thank you for the suggestion. **We have now added these new results to Appendix C.** We plotted the normalized return obtained by REGENT for different choices of context ordering at evaluation time in unseen environments in Figure 10. We evaluated three options: "current", "random-permuted", and "reverse". In the "current" order, the retrieved tuples of (state, previous reward, action) are placed in order of their closeness to the query state $s_t$. In "random-permuted", the retrieved tuples are placed in a randomly chosen order in the context. In "reverse", the "current" order of the retrieved tuples is reversed with the furthest state in the 19 retrieved states placed first. Our current choice performs the best overall. It significantly outperforms the other two orderings in the more challenging task of generalizing to unseen Atari games which have different image observations, dynamics, and rewards. In some metaworld environments on the other hand, all three orderings have similar performance denoting that all the 19 retrieved states for each query state are very similar to each other.
> >
> > > In Figure 15 in Appendix D not all curves show improvement for using more demonstrations, do the authors have an intuition why this is the case?
> >
> > We speculate that in more challenging environments, additional (randomly chosen) demonstrations provide more coverage and hence help improve REGENT’s performance.
> >
> > > From Figures 11, 12, 13, 14 it can be seen that JAT is particularly good on BabyAI, but not good on other domains, while REGENT is very good at Metaworld. Do the authors have an intuition why this is the case?
> >
> > Overall, we observe the REGENT is better than JAT/Gato on the pre-training suites. As the reviewer pointed out, REGENT is much better on metaworld and slightly worse on BabyAI. REGENT is also better than JAT/Gato on Atari and performs similarly on Mujoco. We speculate that this might be because of the larger randomization in babyai tasks.

---

> ### Comment · Reviewer_8qsk · 2024-11-22
>
> **Significance of results:**
>
> Thank you, I greatly appreciate the added standard deviations which address my point about significance of results, which is why I am increasing my score.
> I do not understand though why "there is good reason to expect the standard deviations to be small. ", can the authors elaborate?
>
> **Fine-tuning procedure:**
>
> Regarding my point about fine-tuning, [1] only compares LoRA vs full-finetuning, therefore the claim "Full finetuning has been shown to be much better than **any** parameter-efficient finetuning" is not justified.
> The work I referenced [2] is particularly designed for the few-shot scenario, therefore it might work well in this scenario as well and mitigate the effect of JAT/GATO overfitting.
> Further, it can be easily run (e.g. PEFT library, see https://huggingface.co/docs/peft/v0.13.0/en/conceptual_guides/ia3#ia3) and should be much faster than full-finetuning runs.
> In the few-shot scenario it has shown to be better than LoRA and also full-finetuning.
>
> [1] Shuttleworth et al. LoRA vs Full Fine-tuning: An Illusion of Equivalence, arxiv
>
> [2] Few-Shot Parameter-Efficient Fine-Tuning is Better and Cheaper than In-Context Learning, Liu et al., NeurIPS 2022
>
> **Baselines:**
>
> Thank you for clearing up the question around MTT and adding that information in the appendix.
> The benefits that REGENT does not require re-training are clear to me, I am looking forward to the results on other imitation learning techniques, since I still believe BC is a weak baseline and a comparison to better imitation learning baselines is required to put the results into perspective.
>
> **Related work:**
>
> Thank you for making me aware of the submission date for RA-DT. I appreciate that the authors still added a discussion on that.
> I believe terming REGENT as in-context imitation learning as done by the authors is a good compromise.
>
> **Dependency on order:**
>
> Thank you for conducting this experiment, it confirms my suspicion that the order is essential on most environments.
> I believe it would be good to also briefly higlight this in some discussion section.
>
>
> Thank you for also clearing up the remaining minor points. I am happy to raise my score even further, if the authors can provide a comparison to established imitation learning techniques.

---

> ### Author Response · Authors · 2024-11-24
> **Thank you and Addressing Your Remaining Concerns**
>
> Thank you very much for the quick response and for updating your score. We have addressed your remaining concerns below.
>
> > Significance of results
>
> We are grateful that the reviewer appreciated the standard deviations in the updated Figure 4. **We have now also updated Figure 5 (and the corresponding Table 7) in the paper with the standard deviations for ‘REGENT’ and ‘MTT’ in the ProcGen setting.**
>
> To obtain the standard deviation for ‘REGENT’ in Figure 5 in the ProcGen setting, we trained three REGENT generalist agents for three different seeds. We evaluated each of the three agents’ performance (like before) on each unseen environment across 10 levels with 5 rollouts and sticky probability of 0.2. We computed the final average over both training seeds and evaluation levels/rollouts and the final standard deviation over training seeds. As mentioned in the earlier response, we obtained MTT’s standard deviation along with their best score from their paper.
>
> > I do not understand though why "there is good reason to expect the standard deviations to be small. ", can the authors elaborate?
>
> We simply meant that with the large randomization in evaluation rollouts and the similar performance of generalist policies across training seeds in related work like MTT, we believed, and have now shown, that the standard deviations are small in both settings.
>
> > Fine-tuning procedure
>
> We thank the reviewer very much for this suggestion. **We have now finetuned JAT/Gato and JAT/Gato (All Data) with the IA3 PEFT method and plotted the results in the updated Figure 4 in the paper.** We used the huggingface implementation recommended by the reviewer and the hyperparameters given in the IA3 paper (i.e. 1000 training steps, batch size of 8, and the adafactor optimizer with 3e-3 learning rate and 60 warmup steps in a linear decay schedule).
>
> We found that finetuning JAT/Gato with IA3 improved performance over our simple full finetuning recipe, especially in 3 metaworld environments.
>
> REGENT still outperforms both the IA3 finetuned JAT/Gato and the fully finetuned JAT/Gato in all unseen environments in Figure 4. We have added the above details to Appendix C. We have also updated [a branch of our anonymous codebase](https://github.com/regent-research/regent/tree/more_baselines) to include the code for the above IA3 PEFT method for JAT/Gato.
>
> > Baselines (in Imitation Learning recommended by the reviewer, namely – GAIL [3], Moment Matching [5], and DRIL [4])
>
> **We have now updated Figure 4 with DRIL’s performance on all the atari environments which all have sticky probability = 0.05.**
> Please note that DRIL consists of two stages. The first stage is pre training with a BC loss. In the second stage, the DRIL ensemble policy is allowed to interact in the online environment where it reduces the variance between the members of its ensemble with an online RL algorithm. We run the second online stage for 1M steps or around 1000 episodes.
> REGENT and all the other methods in Figure 4, on the other hand, are completely offline and never interact with any environment for training.
>
> While DRIL improves over the BC baseline, REGENT still outperforms DRIL in all three unseen atari environments.
> We added the above information on DRIL to Appendix C. We have also updated [a branch of our anonymous codebase](https://github.com/regent-research/regent/tree/more_baselines) to include the code for the DRIL reproduction on atari environments with sticky probabilities.
>
> Finally, as mentioned in part 2 of our response above, GAIL [3] fails to perform better than random in atari environments with image observations (see [4]) and the Moment matching methods [5] too are limited to vector observation environments only.
>
> > Dependency on order
>
> > I believe it would be good to also briefly higlight this in some discussion section.
>
> Thank you. **We have now added a “Ablations and Discussions” paragraph in Lines 517-522 in the updated pdf briefly discussing our various ablations’ results.**

---

> > ### Comment · Reviewer_8qsk · 2024-11-24
> >
> > Thank you for providing the additional results and incorporating the suggested changes.
> >
> > For completeness, I would like to encourage the authors to also add variance estimates for the IA3 baselines and DRIL, this should be fesible within a reasonable amount of compute.
> >
> > I think this is good work, therefore I am increasing my score.

---

> > > ### Author Response · Authors · 2024-11-24
> > > **Thank you very much**
> > >
> > > Thank you very much for your quick response, for updating your score, and for supporting our work. We will add the standard deviations for the new IA3 and DRIL baselines as soon as we obtain them.

---

> > > > ### Author Response · Authors · 2024-11-25
> > > > **Quick update**
> > > >
> > > > We have now added the standard deviations for the new IA3 and DRIL baselines. Thank you again.

---

### Official Review · Reviewer_eumA · 2024-11-03

**Soundness:** 2
**Presentation:** 3
**Contribution:** 2
**Rating:** 5
**Confidence:** 3

**Summary:**

The paper proposes REGENT, a retrieval-augmented generalist agent that can adapt to new environments without fine-tuning. REGENT can outperform other RL  agents with fewer parameters in two settings.

**Strengths:**

1. The methodology is well-explained and easy to understand.

2. REGENT demonstrates better performance than larger models (JAT/Gato) with fewer parameters across multiple environment types.

**Weaknesses:**

1. Limited Comparison Scope:

While the paper draws inspiration from retrieval-augmented generation (RAG) in language models, it lacks comparisons with recent LM embodied agents that use retrieval-augmented methods. Some existing work:
- https://arxiv.org/abs/2308.10144
- https://arxiv.org/abs/2402.03610

2. Insufficient Ablation Studies, some valuable ablations could have included:
- Analyzing performance w/wo image information in the context
- Testing various context sizes
 - Analyzing the REGENT's inference latency per state is crucial for real-world deployment. Given the retrieval mechanism and context processing at each step, runtime performance analysis would be valuable for assessing practical applicability


3. Training Data Requirements:

While REGENT shows improved data efficiency compared to baselines like JAT/Gato, it still requires substantial pretraining data across different embodied settings (14.5M transitions in JAT/Gato setting, 12M in ProcGen). This raises questions about its true few-shot learning capabilities and practical applicability in scenarios where large-scale demonstration data is unavailable across task families.

**Questions:**

See weaknesses.

---

> ### Author Response · Authors · 2024-11-20
> **Official Comment by Authors**
>
> We thank the reviewer for the insightful feedback and helpful suggestions. Please let us know if you have lingering questions and whether we can provide any additional clarifications during the discussion period to improve your rating of our paper.
>
> > While the paper draws inspiration from retrieval-augmented generation (RAG) in language models, it lacks comparisons with recent LM embodied agents that use retrieval-augmented methods.
>
> We thank the reviewer for pointing us to these papers (Expel, RAP) that build LLM agents focused on high-level planning. While our work focuses on end-to-end agents that predict low-level actions, these papers highlight the growing recognition of a retrieval bias as a useful mechanism, which aligns with our findings.
>
> We just did an experiment with GPT4 and 4o that showed that it was unable to predict correct/useful low-level actions in the highly continuous environments of metaworld and mujoco even when provided with the same retrieved context as REGENT. The LLM agents in both the suggested papers depend on frozen off-the-shelf LLMs to act as a policy and require existing high level skills as an action space and hence would not be directly applicable to our settings. We believe that these LLM agents are more suitable for higher level control with a semantic text-based action space where they are not required to predict fine-grained low level continuous actions. **We have added the above discussion on these two papers to our extended related work in the newly added Appendix E in the updated pdf.**
>
> > Analyzing performance w/wo image information in the context
>
> We weren’t sure we understood this comment, and guessed that the reviewer was curious about whether / how well REGENT could perform without any context at all. We require at least one retrieved state, reward, action for our interpolation between R&P and the output of the transformer. Our formulation is meant specifically for few-shot (at least 1 shot) in-context learning.
>
> Please do let us know if we have not addressed your question.
>
> > Testing various context sizes
>
> Thank you for the suggestion. **We have now added these new results to Appendix C.** We plotted the normalized return obtained by REGENT for different context sizes at evaluation time in unseen environments in Figure 9. We find that, in general, a larger context size leads to better performance. We also notice that the improvement in performance saturates quickly in many environments. Our choice of 20 states in the context (including the query state) leads to the best performance overall.
>
> > Analyzing the REGENT's inference latency per state is crucial for real-world deployment. Given the retrieval mechanism and context processing at each step, runtime performance analysis would be valuable for assessing practical applicability
>
> Thank you for the suggestion. **We have now added inference runtime value results to Appendix C.** In Figure 11, we plotted the inference runtime values of REGENT's two main components in a metaworld environment: retrieval from demonstrations and forward propagation through the REGENT architecture. We find that retrieval takes between 2 to 8 milliseconds of time (for increasing number of demonstrations to retrieve from) while forward propagation takes around 8 milliseconds uniformly. Both these runtimes are small enough to allow real-time deployment of REGENT on real robots under most common circumstances.
>
> > While REGENT shows improved data efficiency compared to baselines like JAT/Gato, it still requires substantial pretraining data across different embodied settings (14.5M transitions in JAT/Gato setting, 12M in ProcGen). This raises questions about its true few-shot learning capabilities and practical applicability in scenarios where large-scale demonstration data is unavailable across task families.
>
> Thank you for the interesting question. In this paper, we demonstrated that it is possible to learn strong adaptation capabilities in pretraining such that REGENT can adapt to completely new environments with only a few demonstrations: below 16 in atari (please see Table 1), 25-100 in metaworld, and 2-10 in procgen (please see the x axis of Figure 5 and 6). This is not possible with JAT/Gato and other SOTA generalist agents. We believe that this adaptation can be extended to real robots as well. Specifically, REGENT can be first pretrained on simulated robots where it is easier to collect pretraining data and can then be deployed in real world environments with only a few demonstrations for adaptation.

---

> ### Author Response · Authors · 2024-11-22
> **Have we addressed your concerns?**
>
> Dear reviewer eumA,
>
> We thank you again for taking the time and effort to help improve our paper. We believe your key concerns were on adding ablations on context length and details on inference time to the paper. We have added both to Appendix C.
>
> We would be grateful for an opportunity to address any pending concerns you can point us to.
>
> Thank you,
>
> Authors

---

> > ### Author Response · Authors · 2024-11-25
> > **Reminder**
> >
> > Dear reviewer eumA,
> >
> > Since we are at the end of the author-reviewer discussion period, we are again reaching out to ask if our response and new ablations in Appendix C have addressed your concerns. Please let us know if you have lingering questions and whether we can provide any additional clarifications today (the second last day of the discussion period) to improve your rating of our paper.
> >
> > Thank you,
> >
> > Authors

---

> > > ### Comment · Reviewer_eumA · 2024-11-26
> > >
> > > Thank you for providing the detailed responses and results. I have a few follow-up questions:
> > >
> > > - While Section 4.2 describes the distance metrics used (l2 for JAT/Gato setting and SSIM for ProcGen), have you conducted any ablation studies comparing different distance metrics (like cosine distance) for nearest neighbor selection?
> > >
> > > - In Appendix C where you analyze different context ordering options ("current", "random-permuted", and "reverse"), could you clarify whether these ordering experiments were only conducted during evaluation or whether you also pretrained different models with different ordering schemes?
> > >
> > > - Your architecture incorporates RAG from the pretraining stage, which differs from typical LLM approaches that often add RAG during inference. Have you experimented with a variant of REGENT that is pretrained without RAG but uses RAG during inference? This comparison would help isolate whether REGENT's data efficiency stems from architectural benefits during pretraining or primarily from RAG during inference.

---

> ### Author Response · Authors · 2024-11-27
> **Addressing your remaining concerns**
>
> Thank you very much for the response. We have addressed your remaining questions below.
>
> > While Section 4.2 describes the distance metrics used (l2 for JAT/Gato setting and SSIM for ProcGen), have you conducted any ablation studies comparing different distance metrics (like cosine distance) for nearest neighbor selection?
>
> Thank you for the question. **We have now added these new results to Appendix C.** We compared REGENT with cosine distance everywhere (from pretraining to evaluation) against our current version of REGENT with $\ell_2$ distance everywhere in the JAT/Gato setting.
>
> While REGENT with cosine distance performs similarly to REGENT with $\ell_2$ distance in 3 unseen metaworld environments, in the other unseen metaworld environments as well as the more challenging unseen atari environments, REGENT with $\ell_2$ distance significantly outperforms REGENT with cosine distance. We speculate that this is because the magnitude of observations is important with both image embeddings and proprioceptive states and this is only taken into account by the $\ell_2$ distance (not the cosine distance). We added the above discussion to Appendix C. We have also [updated a branch of our anonymous codebase](https://github.com/regent-research/regent/tree/even_more_baselines) with the instructions and code for switching from $\ell_2$ to cosine distance.
>
> Also, as we mentioned in our response to reviewer 8qsk, we had tried dalle based embeddings (instead of resnet18) in the JAT/Gato setting. However, it did not work and produced a REGENT policy that could not generalize better than random to new atari environments. We speculate that the imagenet trained resnet18 provided more general features transferable to atari games than the dalle encoder trained on artistic images.
>
> > In Appendix C where you analyze different context ordering options ("current", "random-permuted", and "reverse"), could you clarify whether these ordering experiments were only conducted during evaluation or whether you also pretrained different models with different ordering schemes?
>
> We experimented with different context orders at evaluation time in this ablation.
>
> > Your architecture incorporates RAG from the pretraining stage, which differs from typical LLM approaches that often add RAG during inference. Have you experimented with a variant of REGENT that is pretrained without RAG but uses RAG during inference? This comparison would help isolate whether REGENT's data efficiency stems from architectural benefits during pretraining or primarily from RAG during inference.
>
> Thank you for the suggestion. **We have now added new results for ‘JAT/Gato + RAG (at inference)’ and ‘JAT/Gato (All Data) + RAG (at inference)’ in Figure 4 (and Table 1 in the Appendix).** Please note that JAT/Gato is like a typical LLM— it has a causal transformer architecture and undergoes vanilla pretraining on consecutive states-rewards-actions. In both these RAG baselines, we use the same retrieval mechanism as REGENT. We provide the retrieved states-rewards-actions context to these RAG baselines replacing the context of the past few states-rewards-actions in vanilla JAT/Gato baselines.
>
> REGENT significantly outperforms both JAT/Gato + RAG baselines in Figure 4. This demonstrates the advantage of REGENT’s architecture and retrieval augmented pretraining over just performing RAG at inference time.
>
> But, the JAT/Gato + RAG baselines outperform vanilla JAT/Gato as well as the JAT/Gato fully Finetuned baselines in the unseen metaworld environments. From this we can conclude that RAG at inference time is also important for performance. We have added the above to Appendix A. We have also [updated a branch of our anonymous codebase](https://github.com/regent-research/regent/tree/even_more_baselines) with the code for JAT/Gato + RAG.

---

> > ### Author Response · Authors · 2024-12-02
> > **Reminder**
> >
> > Dear reviewer eumA,
> >
> > We thank you again for taking the time and effort to help improve our paper.
> >
> > Since we are now at the end of the extended author-reviewer discussion period, we are reaching out to ask if our response, new distance metric ablations in Appendix C, and the comparison to a new inference time RAG baseline in Figure 4 have addressed your remaining concerns.
> >
> > Please let us know if you have lingering questions and whether we can provide any additional clarifications today (the last day of the extended discussion period) to improve your rating of our paper.
> >
> > Thank you,
> >
> > Authors

---

### Official Review · Reviewer_N77b · 2024-11-04

**Soundness:** 3
**Presentation:** 3
**Contribution:** 3
**Rating:** 8
**Confidence:** 4

**Summary:**

This paper presents a method that pre-trained on diverse demonstrations across different environments and tasks that can be adapted to unseen environments via in-context learning, with very few expert demonstrations, and without any fine-tuning. The method is a semi-parametric model that leverages a simple retrieval-based 1-nearest neighbor agent and in-context learning based on other retrieved neighbors to generalize to unseen robotics and game-playing environments. The authors show that their approach is more sample-efficient and parameter-efficient than strong baselines such as Gato.

**Strengths:**

+ This paper proposes a novel and interesting idea of combining a 1-nearest neighbor agent with a Transformer pre-trained to perform in-context learning from retrieved N-nearest neighbor demos. This approach seems to work well for both the Gato setup and the ProcGen setup regarding generalization over unseen environments for robotics or game-playing tasks.
+ The results of the simple 1-nearest neighbor agent show that it serves as a surprisingly strong baseline, which provides insights for solving tasks in the unseen environment.

**Weaknesses:**

+ The paper will be stronger if its experiments are also performed on a larger set of manipulation tasks (ManiSkill2, RLBench, etc.)

**Questions:**

+ I am curious what are the results of REGENT without the help of the 1-nearest neighbor agent.

---

> ### Author Response · Authors · 2024-11-20
> **Official Comment by Authors**
>
> We thank the reviewer for supporting our work, for the insightful feedback, and helpful suggestions. Please let us know if you have lingering questions and whether we can provide any additional clarifications during the discussion period.
>
> > The paper will be stronger if its experiments are also performed on a larger set of manipulation tasks (ManiSkill2, RLBench, etc.)
>
> Thank you for the great suggestion. Our current settings were primarily chosen for a fair comparison with JAT/Gato and MTT. We highlight that in the JAT/Gato setting, we have 145 different training environments. We hope to add even more environments (both training and unseen) from ManiSkill2 and RLBench when scaling up REGENT even beyond its current scale in future work.
>
> > I am curious what are the results of REGENT without the help of the 1-nearest neighbor agent.
>
> Thank you for the great question. Without the interpolation between R&P and the output of the transformer, REGENT does not generalize to unseen environments (i.e., it performs like a random agent on unseen environments). This interpolation is key to REGENT. **We added this to Section 6 in the updated pdf in Lines 429-431.**

---

> > ### Comment · Reviewer_N77b · 2024-11-26
> > **Thanks for the author response**
> >
> > The author response has addressed my concern.

---

> > > ### Author Response · Authors · 2024-12-02
> > > **Thank you**
> > >
> > > Thank you for the response and for supporting our work.

---

### Official Review · Reviewer_jvNh · 2024-11-07

**Soundness:** 3
**Presentation:** 2
**Contribution:** 3
**Rating:** 6
**Confidence:** 3

**Summary:**

This work introduces REGENT, a retrieval-augmented generalist agent designed to adapt to unseen environments without finetuning. It leverages in-context learning and achieves significant generalization across different environments, including robotics and game-playing. It proposed a retrieve-and-play method and incorporated a transformer-based policy to train on sequences of queries and retrieved neighbors.

**Strengths:**

1. The work is well-organized and easy to follow.
2. The observation that the simple retrieve-and-play method can match or surpass the performance of state-of-the-art generalist agents in unseen environments is interesting.

**Weaknesses:**

1. This paper uses nearest neighbor retrieval, claiming it is effective for small training datasets. Yet, they use transformer architecture, known as data hungry. These contradict each other. To prove that nearest neighbor retrieval is indeed effective, it should also be compared to other retrieval methods.
2. There are some places that the author overclaims:

- The author claims multiple times that Gato "struggles to achieve transfer to an unseen Atari game even after finetuning, irrespective of the pretraining data". Note that in Section 5.2 and Figure 9, Gato did a comprehensive study to show that they can transfer to new tasks with few demonstrations.

- The author claims that their model has the ability to adapt to new tasks via in-context learning. However, the ICL refers to a model's capacity to learn and perform tasks by observing examples "without requiring additional training." The LLM can do ICL **without training via RAG**. This is conceptually different from what the author claimed ICL in their paper.

- The author claims that methods like decision transformers cannot generalize to new goals caused by changes in visual observation/available control/game dynamics. Nevertheless, the experimental setting in this paper is simple: the simulation benchmark, like Metaworld, does not involve many visual changes between different tasks. To verify that the REGENT can indeed handle these environmental changes, it is recommended that a real robot experiment be conducted, as Gato does in their paper.

3. Why is JAT/Gato finetuned on new demonstrations, as it is already trained by all data, such as 50 tasks in MetaWorld? This seems to make JAT/Gato overfitting on the target dataset, thus performing extremely badly in experiments in Figure 4. The comparisons are not fair, and the author should provide proper reasons and compare them with the vanilla JAT/Gato.

4. Missing highly-related reference [1,2].

[1] Retrieval-Augmented Embodied Agents, CVPR24
[2] Re-ViLM: Retrieval-Augmented Visual Language Model for Zero and Few-Shot Image Captioning, EMNLP

**Questions:**

Please see weaknesses part.

---

> ### Author Response · Authors · 2024-11-20
> **Official Comment by Authors (Part 1 of 2)**
>
> We thank the reviewer for the insightful feedback and helpful suggestions. Please let us know if you have lingering questions and whether we can provide any additional clarifications during the discussion period to improve your rating of our paper.
>
> > This paper uses nearest neighbor retrieval, claiming it is effective for small training datasets. Yet, they use transformer architecture, known as data hungry. These contradict each other. To prove that nearest neighbor retrieval is indeed effective, it should also be compared to other retrieval methods.
>
> We would like to clarify that the 1 nearest neighbor method is just a simple baseline that does not use any of the pretraining environments in any way. REGENT builds on top of this simple baseline, trains the transformer on the pretrained environments, and improves significantly over the simple baseline as seen in our results. Effectively, REGENT is using the pretraining data to get a much more effective and more adaptive agent at deployment in unseen environments.
>
> Moreover, our results which already show that R&P is better than other SOTA generalist agents (but not as good as REGENT) substantiates our claim that retrieval is a useful bias for adaptation. We also appreciate your suggestion to explore other retrieval methods. Could you recommend specific methods that you believe would provide the most insightful comparisons?
>
> Finally, we note that REGENT is orthogonal to retrieval techniques, because REGENT trains a transformer to effectively use the retrieved information. We believe that better retrieval will enable better REGENT performance as well.
>
> > The author claims multiple times that Gato "struggles to achieve transfer to an unseen Atari game even after finetuning, irrespective of the pretraining data". Note that in Section 5.2 and Figure 9, Gato did a comprehensive study to show that they can transfer to new tasks with few demonstrations.
>
> Thank you for the question. We show in our results that JAT/Gato (the open source reproduction of Gato) fails to meaningfully improve after finetuning. This can be seen by the dashed red and orange curves in Figure 4. We refer the reviewer to this figure.
>
> The finetuning results in the original closed-source Gato paper's Figure 9 (that the reviewer mentions)  involved finetuning on 1000s of demos before Gato could adapt (compared to the 10s that REGENT needs in Fig 4) to reach close to expert performance in an unseen metaworld environment. In unseen atari environments as well, REGENT finetunes on about 3 - 10 demos (please see Figure 4 and Table 1 for the conversion from number of states to demos in atari) and achieves up to 80% of expert performance in atari-mspacman while the Gato paper again needs 1000s of demos for their one held out atari environment (atari-boxing).
>
> Moreover, in the same unseen atari-boxing environment in the original closed-source Gato paper, their train from scratch baseline outperforms Gato. We believe this too demonstrates a significant limitation in Gato's ability to adapt efficiently. Hence, we believe our claim is substantiated.
>
>
> > The author claims that their model has the ability to adapt to new tasks via in-context learning. However, the ICL refers to a model's capacity to learn and perform tasks by observing examples "without requiring additional training." The LLM can do ICL without training via RAG. This is conceptually different from what the author claimed ICL in their paper.
>
> **We do not do any additional training in any of our unseen environments for REGENT.** We only do retrieval-augmented training on pre-training environments. Please see Figures 1 and 2. **This means we are generalizing via in-context learning (ICL) to unseen environments.** Yes, in our approach, the transformer is learning to do ICL on many pretraining environments. Similar ideas of “Learning to do ICL” have been applied to LLMs where they too use “in-context learning” to describe the model’s ability to generalize to new tasks [Min et al], [Chen et al].
>
> [Min et al] S Min, M Lewis, L Zettlemoyer, H Hajishirzi. MetaICL: Learning to learn in context. NAACL 22.
>
> [Chen et al] Y Chen, R Zhong, S Zha, G Karypis, H He. Meta-learning via Language Model In-context Tuning. ACL 22.

---

> ### Author Response · Authors · 2024-11-20
> **Official Comment by Authors (Part 2 of 2)**
>
> > (added Nov 21) The author claims that methods like decision transformers cannot generalize to new goals caused by changes in visual observation/available control/game dynamics. Nevertheless, the experimental setting in this paper is simple: the simulation benchmark, like Metaworld, does not involve many visual changes between different tasks. To verify that the REGENT can indeed handle these environmental changes, it is recommended that a real robot experiment be conducted, as Gato does in their paper.
>
> We respectfully disagree. We showed that REGENT can generalize in-context to unseen Atari and unseen Procgen environments as well, which have large environment changes (including visual appearances) from the pre-training environments. Please see Figures 1 and 2 to visually observe the differences between the pretraining games and held-out unseen games in both settings. Each new game here has completely different visual observations, game dynamics, rewards, and a different subset of controls. As we noted in our related work, this is not possible with in-context RL methods like Decision Pretrained Transformer and Algorithm Distillation which can only generalize to new tasks within the same environment (like a new goal position) according to their papers.
>
> > Why is JAT/Gato finetuned on new demonstrations, as it is already trained by all data, such as 50 tasks in MetaWorld? This seems to make JAT/Gato overfitting on the target dataset, thus performing extremely badly in experiments in Figure 4. The comparisons are not fair, and the author should provide proper reasons and compare them with the vanilla JAT/Gato.
>
> No, this is just a misunderstanding. We trained JAT/Gato from scratch leaving out the unseen environments shown in Figure 1 (see Lines 371-372 in the updated pdf and Lines 404-405 in the initial pdf in Section 6 where we described that JAT is trained on the same dataset as REGENT). That is, we trained JAT/Gato on 145 environments (45 metaworld, 52 atari, 9 mujoco, 39 babyai) leaving out the 12 unseen environments (5 metaworld, 5 atari, 2 mujoco). We also direct the reviewer to Figure 4 which includes the performance of the above JAT/Gato policy without any finetuning as well. We hope that this clarifies that no overtraining of any sort was performed during finetuning and that the results before finetuning were already present in the plot.
>
> > Missing highly-related reference [1,2].
>
> Thank you for the references. RAEA [1] performs behavior cloning in each new environment with access to a “policy memory bank” and focuses on improving performance in a training task. In RAEA’s appendix, the authors demonstrate very initial signs of generalization to new tasks only after training on a few demonstrations from the new tasks (RAEA’s code is also not publicly available). REGENT on the other hand pretrains a generalist policy that can generalize without any finetuning to completely new environments with different observations, dynamics, and rewards.
>
> Re-ViLM [2] trains a retrieval-augmented image captioning model, which while demonstrating the usefulness of a retrieval bias beyond robotics and game-playing tasks, is not applicable to our settings. **We have added the above discussion to the extended related work in the newly added Appendix E in our updated pdf.**

---

> > ### Author Response · Authors · 2024-11-22
> > **Have we addressed your concerns?**
> >
> > Dear reviewer jvNh,
> >
> > We thank you again for taking the time and effort to help improve our paper. We believe your key concerns were on justifying a few of our claims and on discussing a couple of related papers. We believe we have addressed your concerns by substantiating our claims and resolving misunderstandings. We have also discussed the mentioned related works in the newly added Appendix E.
> >
> > We would be grateful for an opportunity to address any pending concerns you can point us to.
> >
> > Thank you,
> >
> > Authors

---

> > > ### Author Response · Authors · 2024-11-25
> > > **Reminder**
> > >
> > > Dear reviewer jvNh,
> > >
> > > Since we are at the end of the author-reviewer discussion period, we are again reaching out to ask if our response and new results have addressed your concerns. Please let us know if you have lingering questions and whether we can provide any additional clarifications today (the second last day of the discussion period) to improve your rating of our paper.
> > >
> > > Thank you,
> > >
> > > Authors

---

> > > > ### Author Response · Authors · 2024-12-02
> > > > **Reminder**
> > > >
> > > > Dear reviewer jvNh,
> > > >
> > > > Since we are at the end of the extended author-reviewer discussion period, we are again reaching out to ask if our response and new results have addressed your concerns. Please let us know if you have lingering questions and whether we can provide any additional clarifications today (the last day of the extended discussion period) to improve your rating of our paper.
> > > >
> > > > Thank you,
> > > >
> > > > Authors

---

> > > > > ### Comment · Reviewer_jvNh · 2024-12-02
> > > > >
> > > > > Thank you for your efforts. It addressed most of my concerns, and I have increased the score.

---

> > > > > > ### Author Response · Authors · 2024-12-02
> > > > > > **Thank you**
> > > > > >
> > > > > > Thank you very much for your response and for updating your score.

---

### Author Response · Authors · 2024-11-20
**Official Global Comment by Authors**

Dear reviewers,

We thank you all for your insightful feedback and helpful suggestions.

We are grateful that reviewers jvNh, N77b, 8qsk rated our soundness and contribution as good (3). We are happy that reviewers N77b and eumA also rated our presentation as good (3). We are happy to see the many positive comments on our novelty, architecture, efficiency, performance, and theoretical guarantees.

We have addressed each weakness and question for all reviewers below their review. We have also made the following major changes to the paper and highlighted all changes (major or minor) in red in the pdf.

* New results in Appendix C on the effect of number of states/rewards/actions in the context on REGENT’s performance [reviewers eumA and 8qsk]
* New results in Appendix C on the effect of the ordering of the context on REGENT’s performance [reviewer 8qsk]
* Inference runtime values in Appendix C [reviewer eumA]
* Standard deviations and means computed over training seeds and evaluation rollout seeds in Figure 4 which required pre-training 3 base models and then training 96 finetuned models [reviewer 8qsk]
* Simpler and clearer exposition in subsections 4.1 and the start of subsection 4.2 [reviewer 8qsk]
* Additional related work in Appendix E [reviewers jvNh, eumA, 8qsk]

Please let us know if you have lingering questions and whether we can provide any additional clarifications during the discussion period to improve your rating of our paper.

Thanks,

Authors

---

> ### Author Response · Authors · 2024-11-24
> **Update to Official Global Comment by Authors**
>
> Dear reviewers,
>
> We have now also added the following to the updated pdf.
>
> * Comparisons with JAT/Gato after parameter efficient finetuning (PEFT) with IA3 [reviewer 8qsk]
> * Comparisons with another imitation learning method (DRIL) [reviewer 8qsk]
> * A brief “Ablations and Discussions” paragraph in Lines 517-522 [reviewer eumA and 8qsk]
> * Standard deviations computed over training seeds in Figure 5 for ProcGen setting [reviewer 8qsk]
>
> Please let us know if you have lingering questions and whether we can provide any additional clarifications during the discussion period to improve your rating of our paper.
>
> Thanks,
>
> Authors

---

> > ### Author Response · Authors · 2024-11-27
> > **Another Update to the Official Global Comment by Authors**
> >
> > Dear reviewers,
> >
> > We have now also added the following to the updated pdf.
> >
> > * Comparisons to JAT/Gato with RAG at inference time in Figure 4 [reviewer eumA]
> > * Ablations on the distance metric ($\ell_2$ vs cosine) in Appendix C [reviewer eumA]
> >
> > Please let us know if you have lingering questions and whether we can provide any additional clarifications during the discussion period to improve your rating of our paper.
> >
> > Thanks,
> >
> > Authors

---

### Meta-Review · Area_Chair_V68e · 2024-12-21

**Metareview:**

This paper looks at generalist agents that can adapt to new environments via retrieval-augmented policy that retrieves nearby states from demonstrations. This retrieved information is used in-context by the transformer-based policy, and it is shown that retrieval is effective but pre-training with retrieval is even more so. Results are demonstrated across a range of environments including robotics and game-playing. The reviewers appreciated that the paper was well-written and the idea simple and effective. Translation of advancements in large language models such as RAG (retrieval-augmented generation) and in-context learning is indeed an interesting direction. However, a number of concerns were raised about the overall execution, including better comparisons to JAT/Gato, comparison to other retrieval/imitation learning methods, statistical significance, ablations such as context size/information type, and lack of some details (e.g. fine-tuning protocol, etc.). The authors provided an extremely comprehensive rebuttal, providing a large amount of experiments addressing the reviewers, which all mentioned that their concerns have largely been resolved.

 After considering all of the materials, I recommend acceptance of this paper. The  method is simple but interesting, and provides a nice signal of moving towards more generalist agents that can solve new environments. I highly recommend that the authors include many of the updated results and discussions in the full paper, as they make the claims and results much stronger and more rigorous.

**Additional Comments On Reviewer Discussion:**

The reviewers raised a large number of points as mentioned in the meta-reviews, and the authors provided a large number of results that addressed those concerns.

---

### Decision · Program_Chairs · 2025-01-22

Accept (Oral)